# A unifying model to explain frequent SARS-CoV-2 rebound after nirmatrelvir treatment and limited prophylactic efficacy

Shadisadat Esmaeili [1,5] ✉, Katherine Owens [1,5], Jessica Wagoner[2], Stephen J. Polyak[2], Judith M. White [3] & Joshua T. Schiffer [1,4]

In a pivotal trial (EPIC-HR), a 5-day course of oral ritonavir-boosted nirmatrelvir, given early during symptomatic SARS-CoV-2 infection (within three days of symptoms onset), decreased hospitalization and death by 89.1% and nasal viral load by 0.87 log relative to placebo in high-risk individuals. Yet, nirmatrelvir/ritonavir failed as post-exposure prophylaxis in a trial, and frequent viral rebound has been observed in subsequent cohorts. We develop a mathematical model capturing viral-immune dynamics and nirmatrelvir pharmacokinetics that recapitulates viral loads from this and another clinical trial (PLATCOV). Our results suggest that nirmatrelvir's in vivo potency is significantly lower than in vitro assays predict. According to our model, a maximally potent agent would reduce the viral load by approximately 3.5 logs relative to placebo at 5 days. The model identifies that earlier initiation and shorter treatment duration are key predictors of post-treatment rebound. Extension of treatment to 10 days for Omicron variant infection in vaccinated individuals, rather than increasing dose or dosing frequency, is predicted to lower the incidence of viral rebound significantly.

The SARS-CoV-2 main protease inhibitor nirmatrelvir is a drug plagued by contradictions. In a landmark, randomized, double-blinded, placebo-controlled clinical trial with 1364 analyzed individuals, 300 mg of nirmatrelvir boosted with 100 mg ritonavir was given twice daily for five days to high-risk individuals with SARS-CoV-2 infection within 3 days of developing symptoms. Compared to placebo, nirmatrelvir reduced the combined outcome of hospitalization and death by 89%, eliminated death as an outcome, and reduced viral load by 0.87 log after 5 days of treatment[1]. This critical result prompted the Food and Drug Administration (FDA) to issue an Emergency Use Authorization[2]. The drug became the most widely prescribed antiviral for SARS-CoV-2 in the United States, likely preventing thousands of hospitalizations and many deaths[3]. Ritonavir boosted nirmatrelvir was recently licensed by the FDA based on its continued effectiveness and

safety[4] and has outperformed other antivirals in terms of hospitalization and viral load reduction[5].

However, the use of nirmatrelvir/ritonavir in real-world cohorts has identified viral rebound as a significant issue. Viral rebound occurred in 14.2% of individuals in one large cohort and was usually associated with recrudescence of symptoms, though protection against hospitalization and death appeared to be maintained[6] and remains significant despite high rates of population immunity due to vaccination and prior infection[7]. Similar rates of viral rebound were observed between molnupiravir and nirmatrelvir, suggesting the rebound effect is not drug-specific and may pertain to characteristics of SARS-CoV-2 infection and treatment duration[8]. This high incidence of viral rebound exceeded the 2.3% rate observed in the proof-of-concept trial, which did not differ from placebo[9].

[1]Vaccine and Infectious Disease Division, Fred Hutchinson Cancer Center, Seattle, WA, USA. [2]Department of Laboratory Medicine & Pathology, University of Washington, Seattle, WA, USA. [3]Department of Cell Biology, University of Virginia, Charlottesville, VA, USA. [4]Department of Medicine, University of Washington, Seattle, WA, USA. [5]These authors contributed equally: Shadisadat Esmaeili, Katherine Owens. ✉e-mail: sesmaeil@fredhutch.org

Despite its high efficacy as an early symptomatic therapy for high-risk individuals, nirmatrelvir/ritonavir was not authorized for use as post-exposure prophylaxis (PEP). In a clinical trial of post-exposure prophylaxis, nirmatrelvir/ritonavir showed 32% and 37% reductions in symptomatic COVID-19 relative to placebo when given for five or ten days respectively[10]. However, neither of these results reached statistical significance. Notably, molnupiravir, another drug that reduced hospitalization when given during early symptomatic infection, also failed as post-exposure prophylaxis[11]. Only long-acting monoclonal antibodies have demonstrated efficacy for post-exposure prophylaxis[12–14], but these are no longer active against prevalent circulating strains[15].

Early during the COVID-19 pandemic, multiple groups employed mathematical models to simulate the outcomes of clinical trials for SARS-CoV-2[16–22]. These models all accurately predicted that antiviral therapy that was insufficiently potent or given too late during infection might fail to provide clinical benefit[16–19,21]. Our previous modeling results further suggested that viral rebound may occur and was more likely if a drug was dosed during the pre-symptomatic phase of infection when viral loads are still expanding, as occurs in a post-exposure prophylaxis scenario[23]. The proposed mechanism of this effect was that reducing viral load may blunt early immune responses and preserve susceptible cells, allowing viral re-expansion upon cessation of treatment that was of insufficient potency to eliminate all infected cells[24]. The model suggested that this phenomenon could theoretically occur during early symptomatic treatment as well. At the time, we downplayed the significance of model-generated rebound as the phenomenon had yet to be demonstrated clinically. However,

models fit to rebound data now suggest a similar mechanism of action[25].

Here we use an updated model for SARS-CoV-2 viral kinetics that was first validated against a much larger panel of untreated individuals to precisely simulate the virologic outcomes of two nirmatrelvir/ritonavir trials. We identify that the true in vivo potency of nirmatrelvir is significantly less than its in vitro potency, such that drug levels are sub-therapeutic during a portion of the dosing interval. Viral rebound is observed in our simulations and is more likely when the drug is dosed early during infection and is not reduced with a higher dose or dosing frequency. Extended-duration treatment is identified as the best strategy to avoid viral rebound.

## Results
### Viral dynamic, pharmacokinetic, and pharmacodynamic mathematical models

To derive parameters for simulating nasal viral loads in the absence of therapy, we used the mechanistic mathematical model (Fig. 1a) that best recapitulated 1510 SARS-CoV-2 infections in a cohort of 1440 SARS-CoV-2 infected individuals from the National Basketball Association (NBA) cohort[26]. The model assumes a finite number of susceptible cells. An eclipse phase delays viral production by infected cells. In keeping with an early interferon-mediated innate immune response, susceptible cells can become refractory to infection in the presence of infected cells but also revert to a susceptible state at a constant rate. Infected cells are cleared by cytolysis, a constant early immune response rate, and delayed acquired immunity, which is activated in a time-dependent fashion. We used a mixed-effect

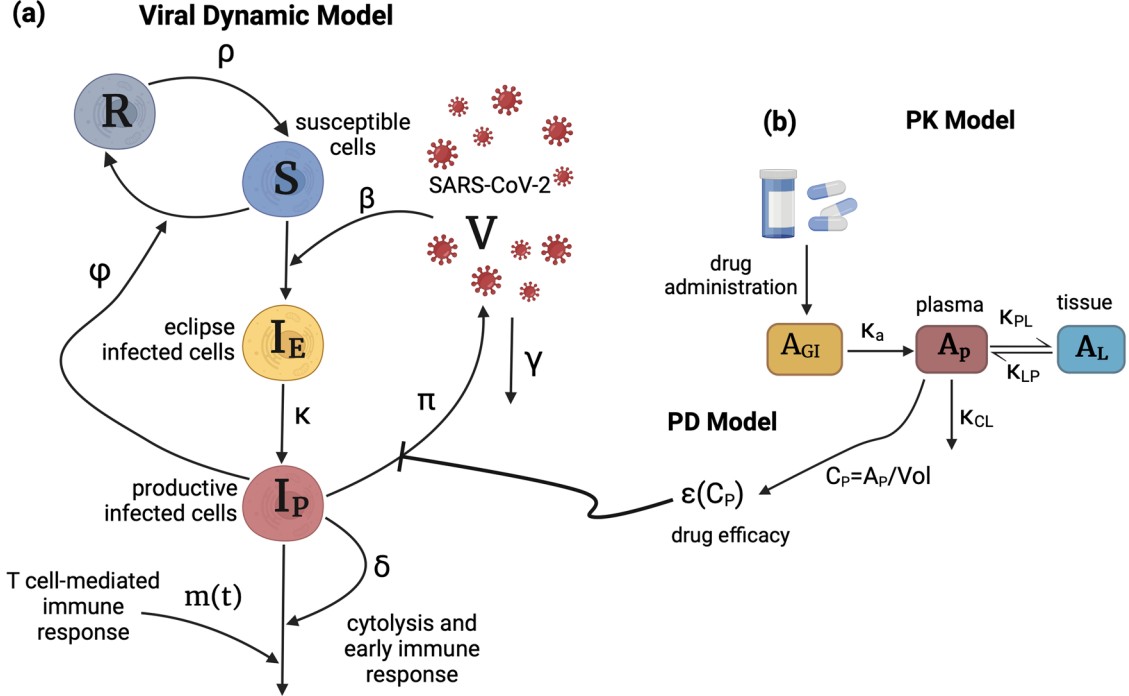

**Fig. 1 | Schematics of the viral dynamic model and nirmatrelvir PK-PD two compartment model. a** The viral dynamic model follows the dynamics of susceptible cells (S), refractory cells (R), eclipse infected cells ($I_E$), productively infected cells ($I_P$), and virus (V) and includes the early and late cytolytic T-cell immune responses with rates $\delta$ and $m(t)$. $\beta$ is the infection rate, $\phi$ is the rate of conversion of susceptible cells to refractory cells, and $\rho$ is the rate of reversion of refractory cells to susceptible cells. Infected cells produce viruses at the rate $\pi$, and the free viruses are cleared at the rate $\gamma$. **b** Two-compartmental PK model with oral administration of the drug which models the amounts of the drug in gut tissue ($A_{GI}$), plasma ($A_P$), and the tissue ($A_L$). $K_a$ is the rate of absorption of the drug from

gut to plasma. $K_{PL}$ and $K_{LP}$ are the rates of transfer of the drug from plasma to the tissue and back, and $K_{CL}$ is the rate at which the drug clears from the body. Vol is the estimated plasma volume and $C_P$ is the drug concentration in plasma. $\epsilon(C_P)$ is the drug efficacy that blocks viral production and is calculated using the Hill equation: $\frac{E_{max} C_P^n}{C_P^n + (prf \cdot IC_{50})^n}$ where Emax is the maximum efficacy, n is the Hill coefficient, $IC_{50}$ is the concentration of drug in vitro at which viral replication rate is reduced by 50%, prf is the potency reduction factor translating the in vitro potency to in vivo potency. (This figure was created with BioRender.com released under a Creative Commons Attribution-NonCommercial-NoDerivs 4.0 International license).

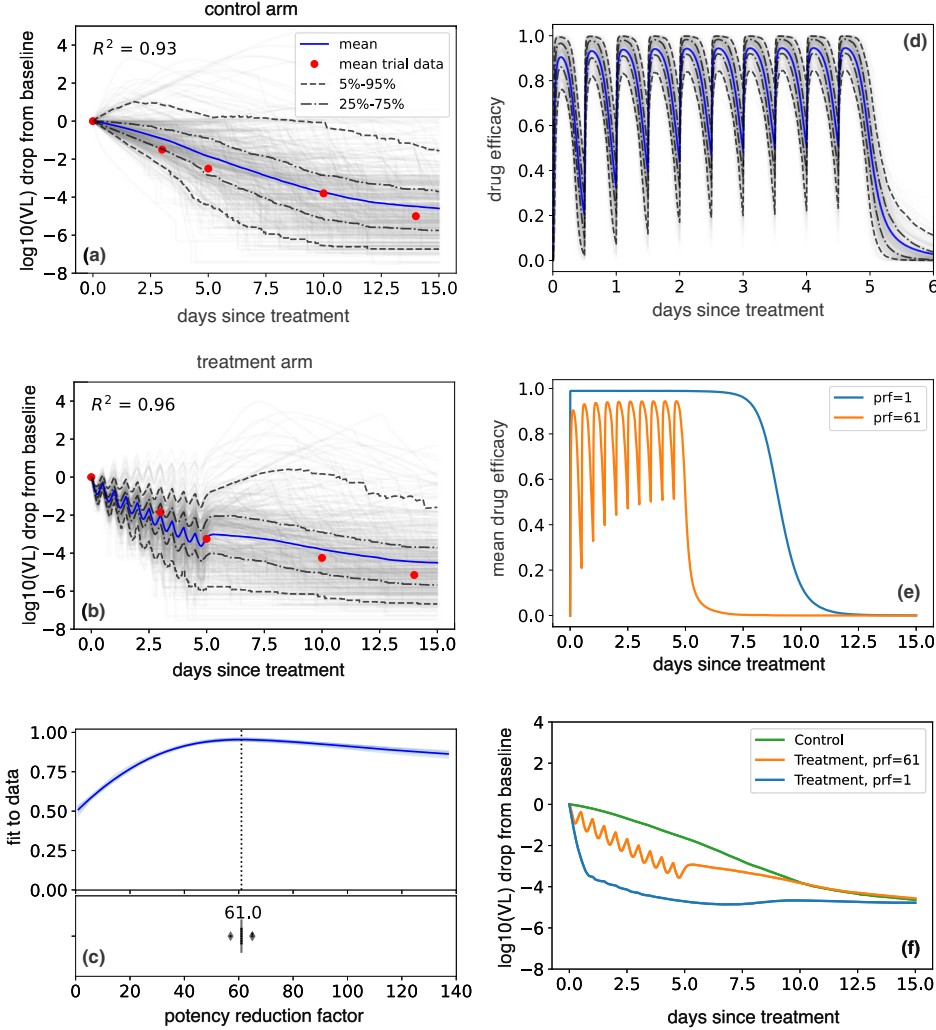

**Fig. 2 | Lower in vivo potency of nirmatrelvir relative to in vitro potency in EPIC-HR.** Mean (blue), individual (gray), and ranges (labeled dashed lines) of log10 viral load drop from the baseline of 400 individuals randomly selected from the NBA cohort treated with (**a**) placebo or (**b**) five days of nirmatrelvir / ritonavir 300 mg twice daily. The red dots were obtained by digitizing Fig. 3a of Hammond et al.[1] and model fit was noted by closeness of blue lines to the red dots. **c** $R^2$ of the fit of the 10 independent model simulations per prf to the viral load drop data in light blue and their mean in dark blue. The best model fit was at a potency reduction factor of 61. The horizontal boxplot in the lower panel shows the distribution of prf

values ($n = 10$) at which $R^2$ is maximum (mean = 61.4, median = 61, sd = 2.15). In the boxplot, the centerline is the median, box limits are the upper and lower quartiles, and whiskers show a 1.5x interquartile range. **d** Drug efficacy when prf = 61. Average efficacy was 82% over the 5-day interval, with notable drops in antiviral efficacy at drug troughs. **e** Projected average drug efficacy when prf = 1 vs prf = 61. The drug with no potency reduction has nearly perfect efficacy (average efficacy of 99.99%) over 5 days and has a prolonged post-treatment effect. **f** Projected mean log10 viral load drop from baseline of the control arm, treatment arm with prf = 61, and treatment arm with prf = 1.

population approach implemented in Monolix to estimate model parameters (Fig S1, Table S1).

To reproduce levels of nirmatrelvir, we used a two-compartment pharmacokinetic (PK) model (Fig. 1b). Using Monolix and the mixed-effect population approach, we estimated parameter values by fitting the model to the plasma concentration of healthy subjects. The model closely recapitulated observed drug levels following a single dose of 250 mg/100 mg of nirmatrelvir/ritonavir (Fig S2, Table S2). The effect of ritonavir as a metabolic inhibitor is accounted for in nirmatrelvir's clearance rate in the PK model. We also fit the model to population-level plasma concentrations following a single dose of 250 mg/100 mg and 750 mg/100 mg, showing that estimated parameters are dose-independent (Table S3).

For the pharmacodynamic (PD) model, we assumed drug efficacy follows a Hill equation with respect to concentration. We parameterized the model using in vitro efficacy data collected at different concentrations of nirmatrelvir (details in "Materials and Methods", Fig S3, Table S4).

Finally, we combined the viral dynamic and PKPD models by using treatment efficacy to lower the viral production rate (details in "Materials and Methods", Fig. 1). We fit the combined model to viral load drop from baseline reported in two randomized, controlled trials: the EPIC-HR trial with 1574 high-risk unvaccinated symptomatic individuals[1] (Fig. 2) and the PLATCOV trial with 144 low-risk, symptomatic individuals (Fig. 3a–e)[5]. We also fit the combined model to individual viral load data from PLATCOV (Fig. 3f–h, Fig S4 & Fig S5, Table S5).

### Mathematical model fitting to clinical trial virologic outcome data

The in vivo potency of a drug is often different from values measured in vitro[23,27,28]. We define the in vivo $IC_{50}$ as the plasma drug concentration required to inhibit viral replication by 50% and the potency reduction factor (prf) as the ratio between the in vivo and in vitro $IC_{50}$. To identify the in vivo potency of nirmatrelvir, we estimated the prf using two methods.

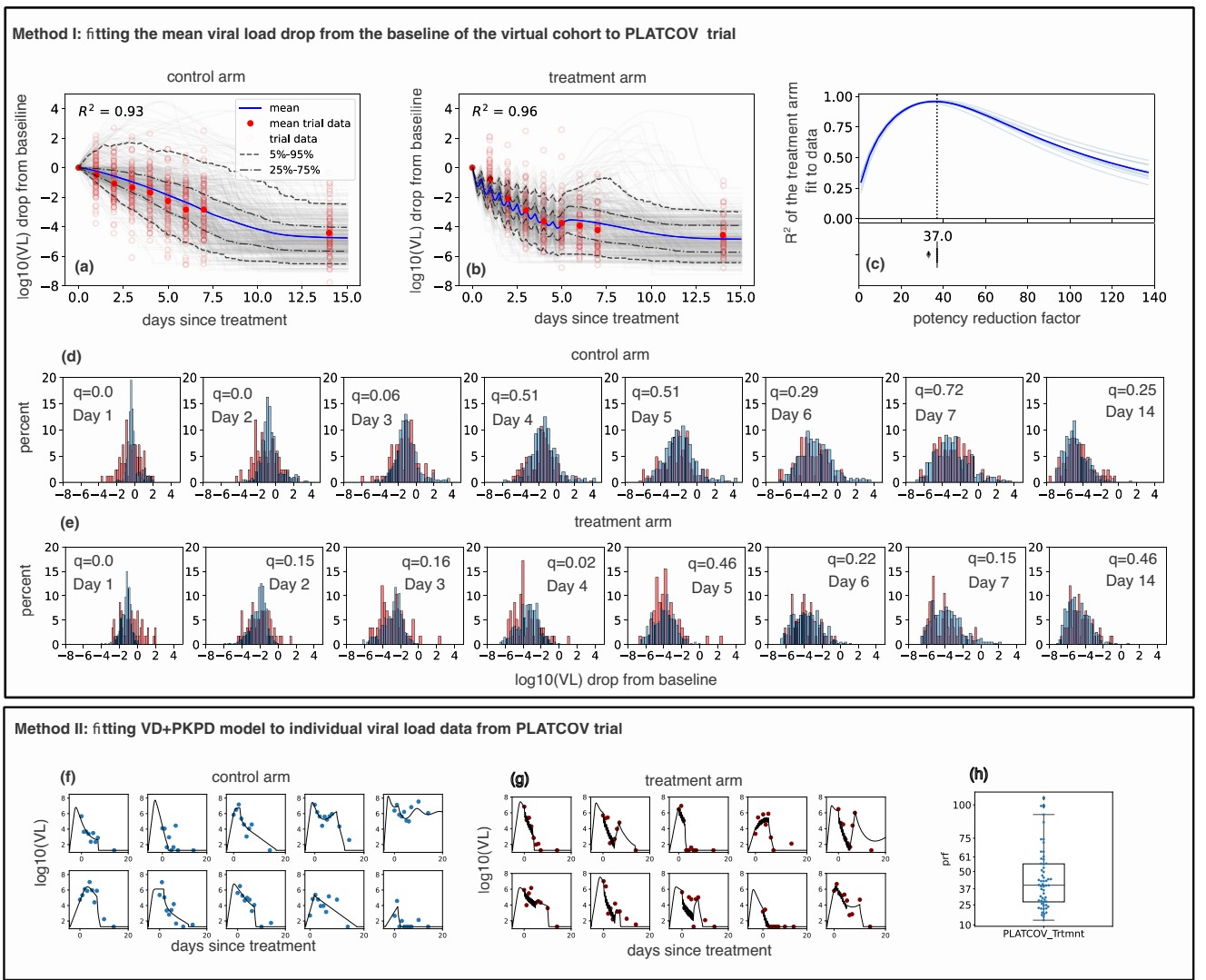

**Fig. 3 | Lower in vivo potency of nirmatrelvir relative to in vitro potency in PLATCOV.** Method I: mean (blue), individual (gray), and ranges (labeled dashed lines) of log10 viral load drop from the baseline of 400 individuals randomly selected from the NBA cohort treated with (**a**) placebo or (**b**) five days of nirmatrelvir / ritonavir 300 mg twice daily. The empty and filled red circles are individual and mean viral load drop from baseline calculated from viral load data published by Schilling et al.[5]. Model fit was noted by closeness of blue lines to the filled red dots. **c** $R^2$ of the fit of the 10 independent model simulations per prf to the viral load drop data in light blue and their mean in dark blue. The best model fit was at a potency reduction factor of 37. The horizontal boxplot in the lower panel shows the distribution of prf values ($n = 10$) at which $R^2$ is maximum (mean = 36.6, median =37,

sd=2.15). In the boxplot, the centerline is the median, box limits are the upper and lower quartiles, and whiskers show a 1.5x interquartile range. Distribution of log10 viral load drop from baseline of simulated cohort and the 144 individuals in PLATCOV control arm (**d**) and treatment arm (**e**). The two-sided Kolmogorov-Smirnov test was used to compare the distributions. Adjusted $p$ values (q-values) were calculated using the Benjamini-Hochberg method and represent dissimilarity between observed and simulated distributions. Method II: sample individual fits to PLATCOV trial participants in control (**f**) and treatment (**g**) arms. **h** Distribution of estimated individual prf values (center line, median; box limits, upper and lower quartiles; whiskers, 1.5x interquartile range, blue dots are the prf values for each individual in the treatment arm, $n = 58$). Remaining fits are in Figs S4, S5.

For the first method, we simulated virtual cohorts using the combined viral dynamic-PKPD model and fit the results to viral load decay from baseline in two trials. For each trial arm, we randomly selected 400 individuals from the NBA cohort with the closest matching viral variant, symptom, and vaccine status (unvaccinated symptomatic subgroup for EPIC-HR and symptomatic Omicron infection for PLATCOV) and used their estimated individual viral dynamic parameters in simulations. This approach generated a wide, realistic range of shedding kinetic patterns among simulated participants (Fig S1).

We next addressed variability in the timing of baseline viral load measurement relative to infection. We randomly assigned all individuals an incubation period selected from a variant-specific gamma distribution found in the literature[29]. Treatment start day was

randomly selected from a uniform distribution for each individual within 3 days of symptoms onset for EPIC-HR trial and within 4 days for the PLATCOV trial.

For all simulated individuals in the treatment arm, PK parameters were randomly drawn from the estimated lognormal population parameter distributions (Table S2) and PD parameters from a normal distribution with estimated mean and standard error (Table S4). To estimate the prf, we simulated our virtual cohort treated with 300 mg of nirmatrelvir twice per day for five days with a range of values and selected the prf that generated the best agreement between the average change from baseline in the treatment arm of each trial and each simulation.

Our simulations recapitulated the mean change in viral load from baseline to multiple timepoints during the two weeks following study

enrollment in EPIC-HR (Fig. 2a) and PLATCOV (Fig. 3a). Similarly, with optimized prf estimates, the model closely recapitulated mean viral load reduction in the treatment arms of both trials (Figs. 2b & 3b).

Our model also predicted individual-level variability in virologic responses observed in PLATCOV, including instances of increased viral load following therapy. We compared simulated and actual distributions of viral load change among trial participants in the control and treatment arms. On most post-treatment days, simulated and actual distributions were not statistically dissimilar (Fig. 3d, e). Wider distributions of observed versus simulated viral load change were noted on post-randomization days 1 and 2 for control and days 1 and 4 for treatment (Fig. 3d, e), perhaps due to noise in viral load data from oral swabs: differences of 1–2 logs were often noted between replicates collected from PLATCOV participants at equivalent timepoints, particularly on day 1 and 2 (Fig S6).

### Reduction of in vivo nirmatrelvir potency relative to in vitro

We plotted the coefficient of determination, $R^2$, for fit to viral load data assuming different prf values (Figs. 2c, 3c). The best values (prf=61 for EPIC-HR and prf=37 for PLATCOV) were determined by maximizing the $R^2$ of the fit. We repeated the simulation 10 times: the boxplot in the lower panel of Figs. 2c, 3c represents the standard error of the prf average value and does not reflect individual variability.

The reason for slight differences in estimated prfs between the two trials is unknown. Possible explanations include different sampling methods (nasal swabs in EPIC-HR versus oropharyngeal swabs in PLATCOV), different trial participant characteristics (high-risk adults in EPIC-HR versus lower-risk adults without comorbidities in PLATCOV), and differing dominant viral variants between the trials.

### Mathematical model fitting to individual viral load trajectories in PLATCOV

For the second method, we fit the combined viral dynamic-PKPD model to individual viral load data from the PLATCOV trial. Since samples were collected after enrollment, we also included data from symptomatic Omicron-infected individuals in the NBA cohort to inform the population model about viral expansion rates during early infection. We used a mixed-effect population approach in Monolix to estimate each participant's viral dynamic parameters and their potency reduction factor (prf) (details in "Materials and Methods"). Our model closely recapitulated viral load trajectories, including cases with post-treatment rebound (Fig. 3f, g). The estimated individual prf values were lognormally distributed, with a median of 39.79 (IQR 27.25–55.75, range 13.51–105.03) (Fig. 3h, Table S5).

The estimated population distribution of viral load parameters for Omicron-infected individuals of the NBA cohort and the PLATCOV trial were the same except for φ (a proxy for the innate immune response), τ (timing of the adaptive immune response), and $t_0$ (infection time) (Fig S7). Time is measured relative to the day of detection in the NBA cohort and relative to the day of baseline measurement in the PLATCOV trial, so the larger $t_0$ and τ values for PLATCOV reflect the delay between infection and trial enrollment. The reason for slight differences in estimated φ values between the two groups is unknown but might be due to the different sampling methods (nasal swabs for NBA versus oropharyngeal swabs for PLATCOV).

To further validate our model, we ran counterfactual simulations switching the PLATCOV treatment and control arms (Fig S8c, d). We treated participants in the control arm of the trial (treatment counterfactual) and removed treatment from participants in the treatment arm (control counterfactual). Due to treatment effect, onset of the adaptive immune response was not easily identifiable for the treatment arm. Therefore, when running the control counterfactual simulation, we assigned random τ values from the estimated control arm distribution. Counterfactual simulations reproduced the mean viral load drop from baseline observed in the trial (Fig S8a, b) and predicted

a diversity of responses to treatment. In some cases, treatment lowered the peak and shortened infection (Fig S8c(I) & d(III)), while in other cases, treatment had a more limited effect (Fig S8c(IV) & d(II)). Our results suggest that some individuals with treatment-induced rebound may not have rebounded in their counterfactual case (Fig S8c(III)), while some untreated individuals with persistent infection might have experienced a treatment-induced rebound (Fig S8d(I)).

### Estimates of viral load reduction with an optimal drug

To illustrate the importance of estimating in vivo drug potency, we compared the PKPD projection and average change in viral load of treatment arms with prf=1 (no reduction in potency) and prf=61 (as estimated in the EPIC-HR trial). With an approximately 61-fold weaker potency, drug levels dropped below therapeutic level shortly after each dose, due to its short half-life (t½), and antiviral effect subsided within a day after treatment ended maintaining an average efficacy of 82% (Eq. (3)) over the first 5 days of treatment (Fig. 2d, e). However, the plasma concentration of a perfectly potent drug (prf=1) remained above therapeutic levels for the duration of treatment with a 5-day average efficacy of 99.99% and the effect persisted for nearly 10 days (Fig. 2e). If the in vivo potency perfectly matched the measured in vitro potency (prf =1), the same treatment regimen could reduce the viral load by approximately 3.5 logs at day 5 relative to the placebo compared to the 0.87 log reduction reported in the trial (Fig. 2f). While estimating nirmatrelvir's in vitro PD parameters, we assumed only the $IC_{50}$ differs in vivo. To confirm the validity of this assumption, we simulated the treatment arm of EPIC-HR with different combinations of the prf and the Hill coefficient. Fig S9 shows that the best fit always happened for prf ~60 and was mostly independent of the Hill coefficient.

The potency reduction factor was more sensitive to certain PK parameters (Fig S10), particularly the drug's clearance rate ($\kappa_{CL}$). If the drug was cleared from the body more rapidly then it would need to be more potent to achieve the effect observed in the clinical trial. However, this did not impact our alternate dosing regimen simulations since PK parameters were independent of the dose (Table S3).

### Frequent viral rebound on nirmatrelvir

To assess whether our model generated viral rebound, we assumed cohort characteristics compatible with the PLATCOV trial (Fig. 3) and randomly drew individual prf values from the distribution obtained by fitting individual data (Fig. 3h, Table S5). We simulated from infection to 30 days after symptoms onset, monitoring viral load continually. We defined rebound in the control arm as any case with at least two peaks in viral load with height greater than 3 logs and higher than its preceding minimum by at least 1 log (Fig S11a). We defined rebound in the treatment arm as any instance in which a post-treatment viral load exceeded the viral load at the end of the treatment by 1 log (Fig S11b).

By this definition, we observed rebound in 18.15% (95% CI [16.64, 19.66]) of cases treated with the clinical trial dose and 1.75% (95% CI [1.23, 2.27]) of controls in our simulations (Fig. 4b). When a less sensitive equivalent definition of rebound was used as in the trial (1 log increase in viral load 5 days after treatment cessation), the probability of rebound in the simulation was much lower (4.12% if treatment was assumed to begin several days after symptoms), closer to that of the controls, and comparable to that observed in the trial (Fig S12).

### Limited impact of nirmatrelvir dose or dosing frequency on viral rebound

We next explored different treatment regimens to estimate their impact on lowering viral load and the chance of rebound. We simulated therapy with 150, 300, 600, and 900 mg doses administered twice per day for 5 days, starting within 3 days post symptoms onset. Larger doses decreased viral load more significantly and quickly than

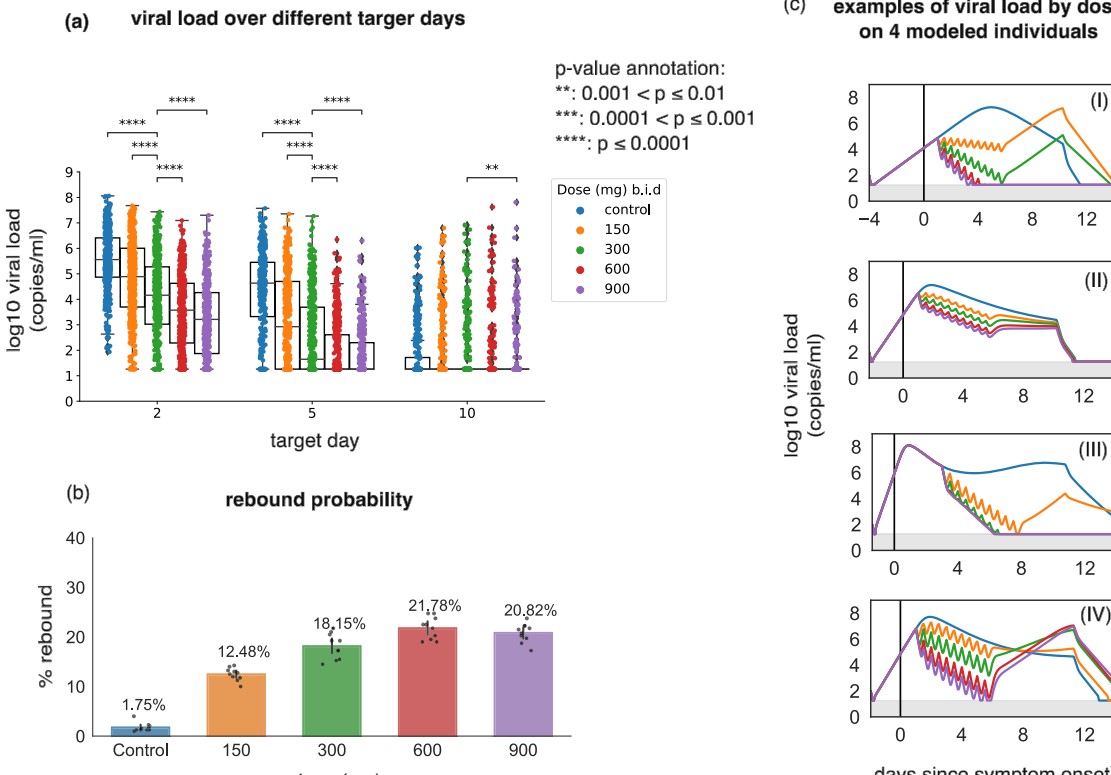

**Fig. 4 | Increasing the nirmatrelvir dose lowers short-term viral load but increases the probability of viral rebound.** In all scenarios, simulated treatment starts within the first 3 days post-symptoms. **a** log10 viral load at days 2, 5, and 10 after the treatment start day with different doses for 400 individuals ($n = 400$). *p* values were obtained by performing two-sided Mann-Whitney U-test between the 300 mg group and the others, and only *p* values < 0.01 are shown. Viral loads were reduced by higher doses at days 2 and 5, but not day 10, except for 900 mg. **b** The probability of rebound for different doses. The error bars on each column are 95% confidence intervals and are obtained by running the simulations 10 times for each dose and the control group. Data points are the percentages obtained from each simulation with the height of each column representing the mean percentage (also annotated above each bar). **c** Examples of viral load trajectories assuming different doses on 4 modeled individuals with equivalent timing of therapy and untreated viral kinetics. In all box plots, the center line is the median; box limits are upper and lower quartiles; whiskers show a 1.5x interquartile range.

300 mg twice daily. 900 mg of nirmatrelvir reduced viral load by a mean of 2 logs on day 2 and a mean of 4 logs on day 5 compared to the control (Fig. 4a).

Individual viral loads were highly variable within each treatment group regardless of dose (Fig. 4a) due to heterogeneous underlying viral dynamics (Fig S1) and different treatment timing. Responses to treatment also differed substantially according to viral load trajectory and treatment timing (Fig. 4c). The reduction in viral load was almost always greater during the first 5 days of treatment with higher doses. However, this only impacted viral elimination in certain cases (Fig. 4c, i). Sometimes, viral load equilibrated to similar levels post-treatment regardless of dose (Fig. 4c, ii), while in other cases, higher doses were associated with rebound (Fig. 4c, iv). By achieving a lower post-treatment viral load nadir, higher doses resulted in a greater likelihood of viral rebound in our simulations (Fig. 4b).

Increasing the frequency of antiviral dosing had nearly equivalent effects to increasing the dose: a more rapid reduction in viral load (Fig S13a), heterogeneous effects based on individual viral dynamics and treatment timing (Fig S13c), and increased chance of rebound (Fig S13b).

**Early treatment as a predictor of SARS-CoV-2 rebound**
We next simulated therapy with four different treatment initiation windows: post-exposure prophylaxis (PEP): 0–1 day after infection in the pre-symptomatic phase; early treatment: 0–1 day after symptoms onset as often occurs in community settings; intermediate treatment: 1–5 days after symptoms onset as in the clinical trial; and late

treatment: 5–10 days after symptoms onset. In all simulations, the administered dosage was 300 mg twice per day for 5 days.

Applying treatment as PEP or shortly after symptoms lowered viral load more substantially relative to control than intermediate or late therapy at days 2 and 5 post-treatment, though intermediate and late strategies also significantly lowered viral load relative to control at these time points (Fig. 5a). However, PEP and early treatment were associated with higher rebound probability after treatment (Fig. 5b, c). The boxplots for control groups in each panel in Fig. 5a show the viral load at different points during infection and are matched to different timing of nirmatrelvir in the treatment arms.

**Prolongation of treatment to reduce the probability of SARS-CoV-2 rebound**
Next, we analyzed the impact of treatment duration on viral rebound. We simulated treatment regimens with 300 mg nirmatrelvir twice per day for 2, 5, 10, 15, and 20 days. Treatment was initiated within 3 days after symptoms appeared. Figure 6a demonstrates the continuous drop in viral load if treatment was maintained until infection was effectively cleared. The viral load distributions of the treatment arms with 15 and 20 days of treatment on days 2, 5, and 10 matched the viral load distribution of the treatment arm with 10 days of treatment duration and, therefore, are not shown. Prolonging treatment duration to 10 days almost eliminated viral rebound (Fig. 6b, c).

We next explored the impact of treatment duration given different treatment initiation times. Prolonging treatment to 15 days for early treatment and 20 days for PEP lowered the viral load close to the

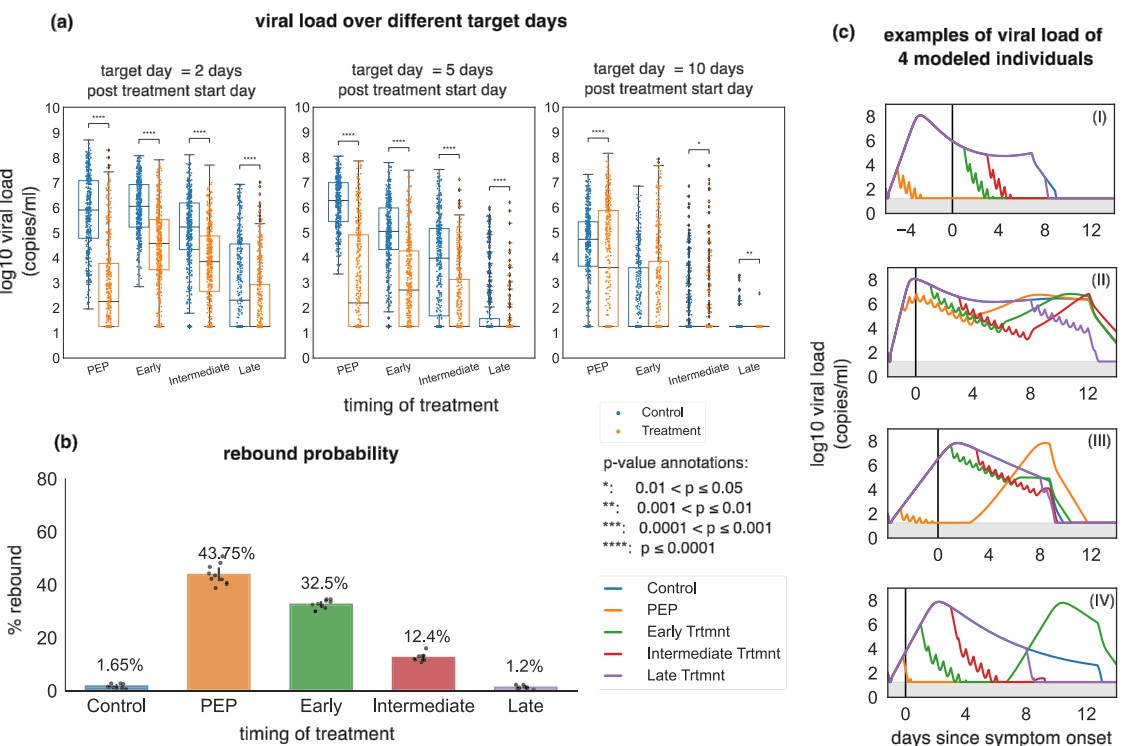

**Fig. 5 | Early timing of therapy initiation is a key risk factor for viral rebound.** In all simulations, the dose was 300 mg twice daily for five days. PEP = 0 to 1 day after infection, early = 1–3 days after symptoms onset, intermediate = 3–5 days after symptoms onset, and late = 5–10 days after symptoms onset. **a** log10 viral load at days 2, 5, and 10 after the treatment start day with different treatment timings for 400 individuals (n = 400). p values were obtained by performing two-sided Mann–Whitney U-test. In all box plots, the center line is the median; box limits are upper and lower quartiles; whiskers show a 1.5x interquartile range. **b** The

probability of rebound for different treatment timings. The error bars on each column are 95% confidence interval and are obtained by running the simulations 10 times for each timing of treatment and the control group. Data points are the percentages obtained from each simulation with the height of each column representing the mean percentage (also annotated above each bar). **c** Samples of viral load trajectories assuming different treatment timings on 4 modeled individuals with equivalent untreated viral kinetics.

---

limit of detection (1.26 log) at the end of treatment and eliminated the probability of rebound for Omicron variants (Fig. 7).

### Differing observed rebound rates resulting from varying timing of sampling and definitions

Previous studies defined rebound using criteria with varying virologic thresholds, timing, and sampling frequency[9]. Rebound was sometimes defined when a positive test followed a negative test[30]. In EPIC-HR, treatment started within 5 days of symptoms onset (our intermediate treatment group) and rebound was defined as a 0.5 log increase on days 10 and/or 14. By this definition 2.3% of treated cases were classified as rebound[9]. The probability of rebound in our simulation with a threshold of 0.5 log measured only on day 5 after the end of the treatment was 5.45% and decreased as thresholds for viral rebound increased (Fig S12). This percentage would be even lower if treatment started 3–5 days after symptoms (rather than 1–5 days) because rebound probability is very sensitive to treatment timing. We hypothesize that in EPIC-HR, participant enrollments skewed later during the 5-day post-symptom window.

In our simulations, we recorded viral load every 0.001 of a day and used a 1 log threshold to identify rebound cases. This was a more sensitive method to observe rebound and suggests that in trial and real-world cohorts, rebound is likely more common in treated individuals than is detected with less frequent sampling (Fig S12).

### Higher rebound probability in unvaccinated individuals with pre-Omicron variant infection

All simulations reported in Figs. 3–7 were performed assuming symptomatic, vaccinated individuals with Omicron infection in the

NBA cohort or PLATCOV. We repeated simulations with characteristics compatible with the EPIC-HR trial (unvaccinated symptomatic individuals with pre-Omicron variants) and prf values randomly drawn from the distribution obtained in Fig. 2c. The same patterns of rebound probability were observed for altered treatment regimens. However, our model predicted an overall higher rebound probability in unvaccinated individuals, infected by pre-Omicron variants (FigS14). While 10 days of treatment would be sufficient to lower the rebound probability significantly in the vaccinated individuals with Omicron infection, 15 days of treatment would have been necessary to substantially lower the incidence of rebound in unvaccinated individuals in the pre-omicron era.

### Immune and viral mechanisms for viral rebound

To understand mechanisms that might explain higher rebound incidence in the PEP and early treatment groups, we simulated four treatment arms with treatment starting on days 1, 4, 7, and 10 after infection. Treatment start relative to infection was fixed to limit the added variability introduced by incubation period and timing of treatment relative to symptoms in previous simulations. High frequency of rebound with day 1 and day 4 treatment start was evident from viral load after treatment (Fig. 8a top row) in many individual trajectories (gray lines) and to a less dramatic extent in mean viral load (blue line). A second peak after treatment ended was also seen in infected cells (Fig. 8a middle row, blue line) and the intensity of the innate immune response (the rate of production of refractory cells) (Fig. 8a bottom row).

Applying treatment earlier during infection (day 1 and 4 in our simulations) lowered the viral load and the populations of infected and

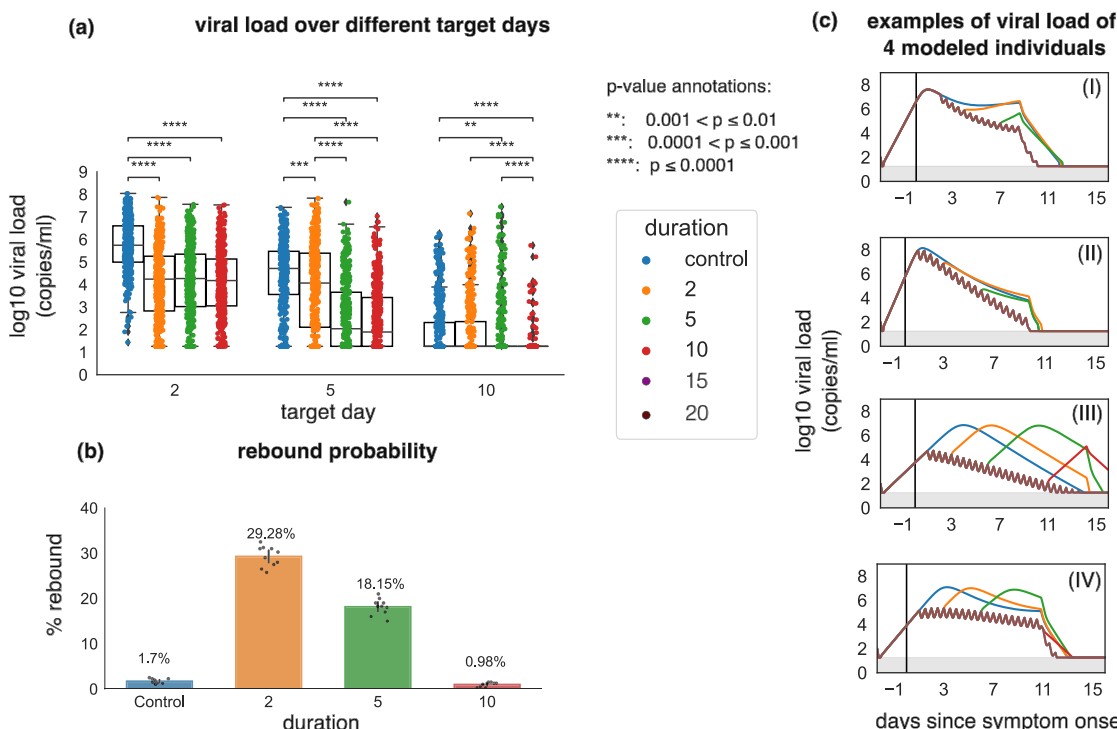

**Fig. 6 | Prolonging treatment duration limits rebound probability.** In all simulations, treatment starts within the first 3 days post-symptoms and the dose was 300 mg twice daily. **a** log10 viral load at days 2, 5, and 10 after the treatment start day with different treatment durations for 400 individuals (*n* = 400). *p* values were obtained by performing two-sided Mann–Whitney U-test and only values < 0.01 are shown. At day 10, the control group had equivalent viral loads to 2 days of treatment while 5 or 10 days of treatment significantly lowered viral load. In all box plots, the center line is the median; box limits are upper and lower quartiles; whiskers show a 1.5x interquartile range. **b** The probability of rebound for different treatment durations. The probabilities of rebound after 15 and 20 days of treatment were zero. The error bars on each column are 95% confidence interval and are obtained by running the simulations 10 times for each duration and the control group. Data points are the percentages obtained from each simulation with the height of each column representing the mean percentage (also annotated above each bar). **c** Samples of viral load trajectories assuming different treatment durations on 4 modeled individuals with equivalent timing of therapy and untreated viral kinetics. Prolonging therapy often avoids rebound.

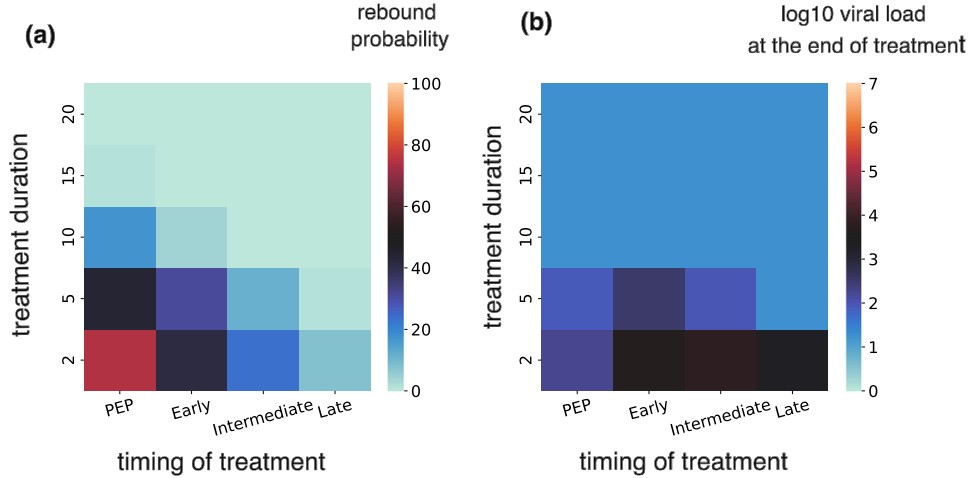

**Fig. 7 | Post-exposure prophylaxis requires more prolonged therapy than early symptomatic therapy to avoid viral rebound. a** Probability of rebound and (**b**) viral load at the end of the treatment as a function of treatment timing and duration.

refractory cells, preserving susceptible cells. In the earlier treatment groups the ratio of susceptible to refractory cells was significantly higher at the end of the treatment than it was in the control group at equivalent time points (Fig. 8b). Innate immune responses were significantly diminished in treated individuals versus controls due to fewer infected cells (Fig. 8c). Overall, a weaker innate immune response, higher availability of susceptible cells, and persistence of

infected cells after 5 days of treatment allowed viral rebound after treatment cessation.

We previously partitioned the NBA cohort according to shedding kinetics using k-means clustering[26]. Groups were ordered by the area under their viral load curve (AUC), with group 1 having the smallest AUC and group 6 the largest (Fig S15a). We simulated treatment with different initiation days using these 6 groups and identified the highest

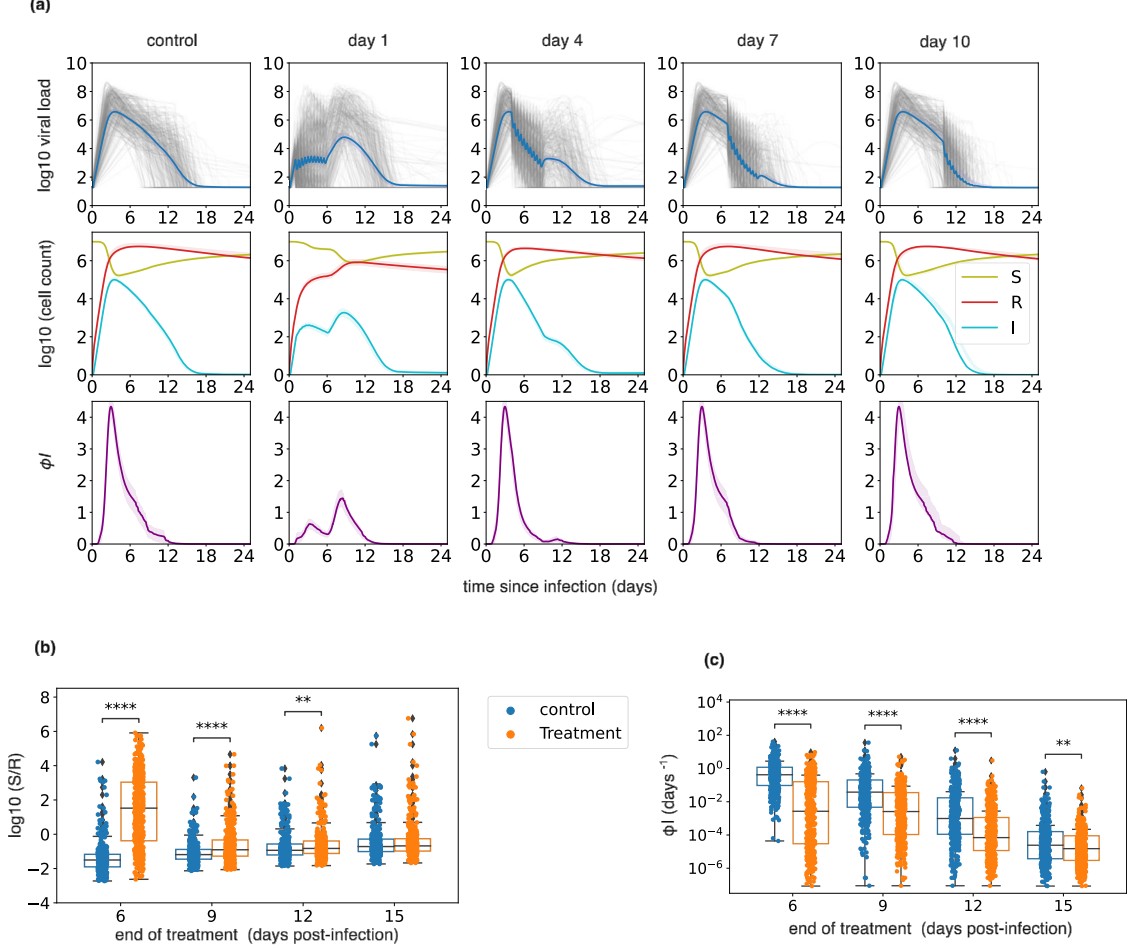

**Fig. 8 | Early therapy preserves susceptible cells, limits refractory cells, does not eliminate all infected cells, and delays innate immune responses.** Simulations are performed using time since infection as a variable rather than based on symptoms as in prior figures to eliminate the confounding impact of variable incubation period. Each group includes 400 individuals ($n = 400$). **a** The top row shows the viral load of all individuals (in gray) and the average viral load (in blue). The middle row shows a less substantial depletion of susceptible cells (S, in green) and a lower generation of refractory cells (R, in red) with earlier therapy. The infected cells (I) are shown in blue. The highlighted area around each line (not visible) shows the 95% confidence interval. The bottom row shows the rate of production of refractory cells, which likely represents innate immune responses per day. The biphasic immune response with a lower initial peak and a second one

after the end of the treatment is observed with early therapy and is present to a lesser extent in day 4 treated individuals. The highlighted areas show 95% confidence intervals. **b** Ratios of susceptible (S) to refractory cells (R) at the end of the 5-day treatment for different timings of treatment ($n = 400$). p-values between control and treatment arms were obtained by performing two-sided Mann–Whitney U-test: $p = 1.1e{-}80$ (day 6), $p = 2.9e{-}12$ (day 9), $p = 2.7e{-}3$ (day 12). **c** Per cell production rate of refractory cells at the end of the 5-day treatment for different timings of treatment ($n = 400$). p-values between control and treatment arms were obtained by performing two-sided Mann–Whitney U-test: $p = 2.3e{-}53$ (day 6), $p = 7.3e{-}27$ (day 9), $p = 6.3e{-}20$ (day 12), $p = 9.0e{-}3$ (day 15). In all box plots, the center line is the median; box limits are upper and lower quartiles; whiskers show a 1.5x interquartile range.

rebound probability in the earlier treatment groups when treating infections that would have fast initial virus expansion (upslope) and high peak viral load (groups 2, 4, and 6) without treatment (Fig S15b, c).

## Discussion

We previously demonstrated for HSV-2[31], HIV[32], Ebola virus[27], and SARS-CoV-2[23], that considering the timing and intensity of the immune response is vital to accurately simulate clinical trials of antiviral agents. If a direct-acting antiviral therapy is given too late during infection, then efficacy is often low because the disease is driven by excess inflammation and cytokine storm. On the other hand, concurrent immune pressure can provide critical assistance for antiviral agents to eliminate viral replication, as confirmed in recent studies[7]. Accordingly, previous modeling suggested that extremely early treatment of pre-symptomatic SARS-CoV-2 as occurs with PEP requires higher drug potency than treatment during early symptomatic infection because innate immunity is more active at this slightly later stage of infection and fewer susceptible cells remain[23]. It is increasingly clear that the

potency and duration of antiviral therapy required to achieve clinical benefit depends strongly on the stage of infection and the ongoing intensity of the immune response.

Prior work also demonstrated that in vitro antiviral drug potency measured in cell culture often overestimates in vivo potency in humans[27,28,33]. Specifically, the plasma drug level required to achieve 50% inhibition of cellular infections in vivo is higher than the level required in vitro. The discrepancy between in vitro and in vivo potency can be assessed by fitting viral dynamic-PKPD mathematical models to viral load data from clinical trials, as we have done here. Traditional PKPD models, which do not include a dynamic immune response, are not sufficient to estimate in vivo potency. Because in vivo potency reduction varies from 2 to 100 depending on the infection and antiviral agents[27,31,33], population in vivo $IC_{50}$ must be assessed separately in each case.

Here, by precisely fitting a combined viral-immune dynamic-PKPD model to viral load data from a randomized clinical trial as well as an open-label clinical trial of nirmatrelvir/ritonavir, we merge these two

key concepts. We first identify that nirmatrelvir potency is reduced 60-70 fold in vivo relative to in vitro in the high-risk population and 30-40 fold in the healthy population. The difference between the estimated in vivo potency in these two populations might be explained by differences in demographics, sampling methods, and the dominant viral variants in the two trials. However, both values fall within the range of inter-individual variability estimated by fitting the model to the individual viral loads of the second trial. The mechanistic reasons for this reduction cannot be determined by the model, but may include increased in vivo protein binding[34], inhibition of drug delivery from plasma to sites of infection, or differences in cellular uptake and drug metabolism in vivo[35]. Nevertheless, our estimated in vivo $IC_{50}$ provides a benchmark plasma level to target in future trials. The PKPD model also demonstrates that the drug's relatively short half-life allows it to dip to subtherapeutic levels even when dosed twice daily.

Our model also develops a viable hypothesis for why nirmatrelvir is highly effective when given during early symptomatic infection but less so when given as post-exposure prophylaxis. By preventing a high peak viral load approximately 3–5 days after infection, therapy preserves susceptible cells and blunts the immediate, likely innate immune response to SARS-CoV-2, while not eliminating infected cells. If the virus is not eliminated by an early acquired response along with antiviral pressure, it rebounds to a peak level that is sometimes comparable to the initial peak. We hypothesize that viral rebound occurs more frequently in community settings relative to the clinical trial, because infected individuals in the community are often prescribed the drug very early after symptom development, whereas in the trial, there was a natural 1 to 2-day delay based on the enrollment and consent process. Surprisingly, this short delay may have limited rebound while not affecting the primary endpoints of the trial, a finding supported by recent clinical studies[30], which nevertheless still suggests a clear benefit for earlier treatment in terms of preventing hospitalization in high-risk individuals[7]. Notably, antiviral therapy is not a risk factor for rebound in our model or in clinical cohorts if administered late during infection[36]. However, high viral load shedding is a risk factor for rebound in our model, as suggested in other studies[6].

Our model identifies optimal conditions for viral rebound, which, counterintuitively, include early treatment during pre-symptomatic infection and higher or more frequent dosing. Both factors suppress the amount of infection, thereby preserving susceptible cells, limiting the development of refractory cells, and dampening the intensity of the early immune response. The best method to prevent viral rebound is prolonging treatment, with a longer course needed for PEP. This finding is consistent with trials of long-acting monoclonal antibodies, which demonstrated efficacy as post-exposure prophylaxis[12–14].

Because our model is validated precisely against mean viral load reduction from two trials as well as individual viral kinetics it can be used as a tool to test treatment strategies with varied therapeutic goals, timing of treatment, dose, dosing interval, and duration of therapy. Our prior PD modeling also allows testing of potentially synergistic agent combinations and consideration of special hosts such as immunocompromised individuals with persistent infection who may be at risk of developing drug resistance[27,37,38]. We believe our approach provides a template for optimizing future trial designs with nirmatrelvir and other therapies.

Our model has several limitations. First, nasal or oropharyngeal viral load is not a perfect surrogate of disease activity. On the one hand, viral load reduction has been correlated with beneficial clinical outcomes for nirmatrelvir[1], molnupiravir[39], and monoclonal antibodies[40]. A recent review shows that viral load reduction is a reasonably good surrogate endpoint[40]. Moreover, viral rebound appears to track very closely with symptomatic rebound in multiple case series[9]. Yet, early remdesivir treatment provided a profound reduction in hospitalization while not impacting nasal viral load, albeit 5 days after completion of therapy[41]. Data from non-human primates suggests that the drug has a specific effect on viral loads in the lungs that is not observed in upper airways, a finding that we were also able to capture with a model[23]. Overall, there is a strong suggestion from early treatment trials that a reduction in nasal viral loads beyond that observed in placebo-treated individuals is associated with substantial clinical benefit[1].

Another limitation is that the model does not account for drug resistance. While there has been limited evidence of de novo resistance during nirmatrelvir therapy, serial passage of virus suggests a relatively low barrier, and some viral rebound could, in theory, be with resistant variants. Studies to date suggest very little mutational change between the infecting and rebounding virus[42–45].

Our model does not capture immunity in literal terms. For instance, we do not distinguish innate interferon, antibody, and T-cell responses, as these have not been measured in sufficient longitudinal detail to precisely ascribe viral clearance to different components of the immune response. We structured the model for the early response to roughly map to innate responses, as the model term capturing the progression of susceptible cells to a refractory state diminishes with decreases in viral load and assumes no immune memory. The late immune response in our model has memory, leads to rapid elimination of the virus, and is likely to represent acquired immunity. While a more accurate model would discriminate different arms of the immune responses and fit to immune data, ours sufficiently captures the timing and intensity of immune responses for accurate clinical trial simulation.

Finally, it is our opinion that models lacking a spatial component cannot fully describe target cell limitation, which is influenced by the packing structure of cells, viral diffusion, and infection within multiple concurrent micro-environments[31]. Consequently, ordinary differential equations may misclassify the relative impact of target cell limitation and innate immune responses in the period surrounding peak viral load. However, our approach provides accurate output for clinical trial simulation.

In conclusion, our model identifies viable mechanistic underpinnings of the high efficacy of nirmatrelvir therapy for early symptomatic SARS-CoV-2 infection, lower efficacy for PEP, and high incidence of viral rebound in a real-world setting. The model can also be used to assess different treatment strategies and suggests prolonging therapy is the optimal method to avoid rebound while maintaining potent early antiviral suppression.

## Methods
### Study design
We developed a viral dynamics model recapitulating viral load data collected from symptomatic individuals in the NBA (National Basketball Association) cohort[46]. We used a two-compartment model to reproduce the PK data of nirmatrelvir plus ritonavir[2]. For clinical trial simulation, we constructed a virtual cohort by randomly selecting 400 individuals from the NBA cohort, trying to match the trial populations regarding vaccine status and history of infection, and assigning individual PK and PD parameters randomly drawn from their respective inferred distributions. We fit the combined viral dynamic and PKPD model to the average change in viral load from the baseline as well as individual viral load data of the control and treatment arms of two previously published nirmatrelvir/ritonavir clinical trials[1,5]. Comparing our model to the control arms validated our viral dynamic model and demonstrated how well our virtual cohorts represent the trial control arms. As one method of fitting the treatment arms, we used the average data from the treatment arms to estimate the potency reduction factor (prf) by maximizing the $R^2$ of the fit. In a second approach, we fit to individual viral load trajectories in PLATCOV using the mixed-effect population approach implemented in Monolix and obtained both

individual prf values and a population distribution. With the estimated prf and in vivo $IC_{50}$ of the drug, we explored different treatment regimens by changing dose, dosing frequency, treatment duration, and treatment timing to find the best strategy to minimize the probability of rebound.

## Viral load data

The NBA cohort dataset published by Hay et al.[46] consists of 2875 documented SARS-CoV-2 infections in 2678 people detected through frequent PCR testing regardless of symptoms. We used the viral load data from 1510 infections in 1440 individuals that had at least 4 positive quantitative samples to estimate viral dynamic parameters. We used parameter sets estimated for the symptomatic subpopulation of these individuals to construct virtual cohorts.

## Clinical trial data

We used viral load data from two nirmatrelvir/ritonavir clinical trials. EPIC-HR by Hammond et al.[1] included 682 and 697 symptomatic high-risk individuals in the control and treatment arms, respectively. We obtained the average change in viral load data of the control and treatment arms by digitizing Fig. 3A of the manuscript by Hammond et al.[1]. Nasal viral load was measured using PCR assay on days 0, 3, 5, 10, and 14 after the treatment start day and adjusted by the baseline viral load. PLATCOV by Schilling et al.[5] is an open-label, randomized, controlled adaptive trial with 85 and 58 symptomatic, young, healthy individuals in the control and nirmatrelvir treatment arms, respectively. The oropharyngeal samples from each participant were collected daily on days 0 through 7 and on day 14 after the treatment start day, and viral load was measured using PCR assay. We used the individual viral load data published by the authors. From PLATCOV, we averaged over the two oral samples collected from each individual and calculated viral load drop from baseline (to use in method 1, Fig. 3) or used the individual-level viral load data (in method 2, Fig. 3). In both trials, the study participants were treated with 300 mg/100 mg nirmatrelvir/ritonavir within three days (EPIC-HR) or four days (PLATCOV) of symptom onset. The treatment was administered twice per day, for five days. We used EPIC-HR's lower limit of detection (LOD = 2 logs imputed as 1 log) in the simulations where we used EPIC-HR cohort characteristics (unvaccinated symptomatic individuals) (Figs. S9, S10, S14). When fitting to PLATCOV cohort characteristics (vaccinated symptomatic individuals with omicron infection) and in all the simulations in the main paper, we used the maximum LOD reported in the published data (~1.26 log).

## PK data

PK data of nirmatrelvir (PF-07321332) with ritonavir was obtained by digitizing Fig. 4 of the drug's Emergency Use Authorization document[2] using WebPlotDigitizer. The data is from a phase I randomized trial by Singh et al.[47] where eight participants (4 fed, 4 fasting) took a single dose of 250 mg/100 mg nirmatrelvir/ritonavir. Drug concentrations in plasma were recorded for 48 h following dosing.

## PD data

The data on drug efficacy came from experiments performed at the University of Washington. The efficacy of nirmatrelvir in the presence of CP-100356 (an efflux inhibitor[48]) was measured against the delta variant of SARS-CoV2 in Calu-3 cells (human lung epithelial). The efflux inhibitor ensures consistent, adequate intracellular levels of drug. Briefly, Calu 3 cells were treated with varying concentrations of nirmatrelvir in the presence of 2uM CP-100356 prior to infection with SARS-CoV-2 (delta isolate) at a multiplicity of infection of 0.01. Antiviral efficacy and cell viability (of non-infected cells treated with drugs) were assessed as described[49]. There were five replicates per condition, pooled from 2 independent technical experimental repeats (one experiment with triplicate conditions, one experiment in duplicate conditions).

## Viral dynamics model

We used our model of SARS-CoV-2 dynamics[26] to model the viral load of symptomatic individuals with SARS-CoV-2 infection. Our model assumes that susceptible cells (S) are infected at rate βVS by SARS-CoV-2 virions. The infected cells go through a non-productive eclipse phase ($I_E$) before producing viruses and transition to becoming productively infected ($I_P$) at rate $\kappa I_E$. When encountering productively infected cells, the susceptible cells become refractory to infection (R) at the rate $\phi I_P S$. Refractory cells revert to a susceptible state at rate $\rho R$. The productively infected cells produce virus at the rate $\pi I_P$ and are cleared at rate $\delta I$ representing cytolysis and the innate immune response that lacks memory and is proportional to the amount of ongoing infection. If the infection persists longer than time τ, then cytotoxic acquired immunity is activated, which is represented in our model by the rate $m I_P$. Finally, free virions are cleared at the rate γ. Of note, this model, previously proposed by Ke et al.[50] was selected against other models in [26] based on superior fit to data and parsimony. The model written as a set of differential equations has the form,

$$\frac{dS}{dt} = -\beta SV - \phi I_P S + \rho R \tag{1a}$$

$$\frac{dR}{dt} = \phi I_P S - \rho R \tag{1b}$$

$$\frac{dI_E}{dt} = \beta SV - \kappa I_E \tag{1c}$$

$$\frac{dI_P}{dt} = \kappa I_E - \delta I_P - m(t) I_P \tag{1d}$$

$$\frac{dV}{dt} = \pi I_P - \gamma V \tag{1e}$$

where

$$\begin{cases} m(t) = 0 & t < \tau \\ m(t) = m & t \geq \tau \end{cases} \tag{1f}$$

To estimate parameter values, we fit the model to viral load data from the NBA cohort using a mixed-effect population approach implemented in Monolix. Details on the model selection and fitting process can be found in Owens et al.[26]. Information about parameter distributions and the error model is provided in Table S1.

We start the simulations with $10^7$ susceptible cells. The initial value of the refractory cells is assumed to be zero since the interferon signaling is not active prior to infection. We further assume there are no infected cells (eclipse or productive) at the beginning of the infection. We fix the level of inoculum ($V_0$) at 97 copies/ml for each individual.

To resolve identifiability issues, we fixed two parameter values, setting the inverse of the eclipse phase duration to $\kappa = 4$, and the rate of clearance of virions to $\gamma = 15$[26].

## PK model

We used a two-compartmental PK model which includes the amount of drug in the GI tract ($A_{GI}$), the plasma compartment ($A_P$), and the lung ($A_L$). The drug is administered orally, passes through the GI tract and gets absorbed into the blood at the rate $\kappa_a$. The drug then transfers from the blood into the peripheral compartment (or the lung) at the rate $\kappa_{PL}$. The metabolized drug transfers back into the plasma at the rate $\kappa_{LP}$ from where it clears from the body at the rate $\kappa_{CL}$. The model

in the form of ordinary differential equations is written as,

$$\frac{dA_{GI}}{dt} = -\kappa_a A_{GI} \tag{2a}$$

$$\frac{dA_P}{dt} = \kappa_a A_{GI} + \kappa_{LP} A_L - (\kappa_{CL} + \kappa_{PL}) A_P \tag{2b}$$

$$\frac{dA_L}{dt} = \kappa_{PL} A_P - \kappa_{LP} A_L \tag{2c}$$

We used Monolix and a mixed-effect population approach to estimate the parameters and their standard deviations. With the initial condition of ($A_{GI}$ = Dose, $A_P = 0$, $A_L = 0$); we fit $C_P = \frac{A_P}{Vol}$ to the plasma concentration data where Vol is the estimated plasma volume. Details on parameter distributions and the error model provided in Table S2.

**PD model**

For the pharmacodynamics model we used Hill equation, $\epsilon(t) = \frac{E_{max} C(t)^n}{C(t)^n + IC_{50}^n}$, where C(t) is the drug concentration in plasma, $E_{max}$ is the maximum efficacy, $n$ is the Hill coefficient, and $IC_{50}$ is the drug concentration in plasma required for 50% efficacy. We used least-squared fitting to obtain the three parameters ($E_{max}$, $n$, and $IC_{50}$) and their standard deviations. The average drug efficacy is measured using,

$$E_{ave} = \frac{1}{t_{start} - t_{end}} \int_{t_{start}}^{t_{end}} \epsilon(t) dt \tag{3}$$

Where $t_{start}$ and $t_{end}$ are the treatment start day and end day, respectively.

**Combined PKPD and VL models**

The plasma concentration of nirmatrelvir obtained from the PK model is used in the PD model to obtain time-dependent efficacy. $\epsilon(t)$, then, is used to reduce viral production rate, $\pi$, with the factor of $(1 - \epsilon(t))$. Equation 1e is written as,

$$\frac{dV}{dt} = (1 - \epsilon(t)) \pi I_P - \gamma V \tag{4}$$

**Fitting the combined model to individual viral load data in the PLATCOV trial**

We used the population mixed-effect approach and Monolix to estimate each individual's viral dynamics parameters and the potency reduction factor (prf). Due to the lack of data from the initial phase of infection in the PLATCOV trial, we include the data from Omicron-infected individuals in the NBA cohort to inform the model about the initial phase of infection. We fixed the PK parameters to the estimated population values (Table S2), and the PD parameters other than IC50 to the in vitro estimated population values (Table S4). We used the study category (NBA vs PLATCOV) as a covariate for $t_0$ (timing of infection) and $\tau$ (timing of the adaptive immune response) since the first recorded positive test is likely much later for the clinical trials. In the NBA study, samples were collected almost daily regardless of symptoms often leading to pre-symptomatic detection, while in the PLATCOV study, the baseline measurement occurred after symptom onset, trial enrollment and consent.

**Construction of a virtual cohort**

To generate a cohort for our simulated clinical trials, we randomly selected 400 individuals (for each arm of the simulated trials) from the unvaccinated symptomatic subpopulation of the NBA cohort for EPIC-HR and vaccinated with Omicron infection for PLATCOV and used their

individual viral load parameters estimated by fitting our viral dynamics model to the data. A sample size of $n = 400$ (out of 822 vaccinated individuals with Omicron infection) was used to mimic a large-scale clinical trial and maintain relatively low overlap between virtual cohorts used in each arm of the simulations and between different simulations. Since the time of symptom onset is not available for all individuals in the NBA data, we randomly drew an incubation period for each individual from gamma distributions with variant-specific parameters estimated by Gamiche et al. [29] The start of treatment relative to symptom onset was randomly selected according to a uniform distribution, except when constructing Fig. 8. The PK parameters of each simulated individual were randomly drawn from lognormal distributions, for which estimated mean and standard deviation were inferred from the PK data. The relevant dose in each scenario was added to the $A_{GI}$ compartment (the absorption equation) of the PK model (Eq. 2a) at each dosing timepoint (t = 0, 0.5, 1, 1.5, …., 4.5 days). For all doses, we used the PK parameter distributions estimated for 250 mg since our analysis showed they are dose-independent. PD parameters were also randomly drawn from a normal distribution with the estimated mean and standard deviation. The standard deviation of the PD parameters represents the accuracy of the assays and not individual variability. The individual potency reduction factors were also drawn from a lognormal distribution with estimated mean and standard deviation obtained from fitting the model to the individual viral load data of PLATCOV study.

**Potency reduction factor (prf)**

The prf is defined as,

$$prf = \frac{IC_{50, in\, vivo}}{IC_{50, in\, vitro}} \tag{5}$$

We estimated the prf by maximizing $R^2$ when fitting the change in viral load of the treatment arm of our simulation to the data from the treatment arm of the clinical trial.

**Measuring rebound probability**

A viral load rebound in the treatment arm was defined when the viral load at any time after treatment ended exceeded the viral load at the end of the treatment by 1 log. In the control group, viral rebound was defined in patients who had at least two peaks with maximum height of 1000 copies/ml in their viral load trajectories and the second peak was 1 log higher than its preceding local minimum (Fig S7).

**Software**

All the model fittings in this study were performed using Monolix version 2023R1.

The data analysis and simulations were performed using Python 3.9.12. WebPlotDigitizer (https://automeris.io/wpd/) was used to digitize data.

**Reporting summary**

Further information on research design is available in the Nature Portfolio Reporting Summary linked to this article.

## Data availability

The data analyzed in this work was previously published by Hay et al. and Schilling et al. and is available on GitHub at https://github.com/gradlab/SC2-kinetics-immune-history and https://github.com/jwatowatson/PLATCOV-Molnupiravir/tree/V1.0 Pharmacodynamics data is available on GitHub at https://github.com/sEsmaeili/Covid_Rebound.

## Code availability

All codes and materials used in the analysis are available on GitHub at https://github.com/sEsmaeili/Covid_Rebound[51].

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

## Author contributions

Conceptualization: JTS, SE, KO. Methodology: JTS, SE, KO. Software: SE, KO. Investigation: SJP, JW, SE, KO. Formal analysis: SE, KO. Writing—original draft: JTS, SE. Writing—review & editing: JTS, SE, KO, SJP, JW, JMW.

## Funding

National Institutes of Health (NIH) grants RO1AI169427 (JTS). National Institutes of Health (NIH) grants RO1AI121129 (JTS). National Institutes of Health (NIH) grants RO1AI177512 (JTS, SJP)

## Competing interests

The authors declare no competing interests.
