## [Peer Review File · Nature Communications]

A unifying model to explain high nirmatrelvir therapeutic efficacy against SARS-CoV-2, despite low post-exposure prophylaxis efficacy and frequent viral reboundReviewers' Comments:

Reviewer #1:

Remarks to the Author:

Esmaeili-Wellman et al. developed PK/PD and viral dynamic models to understand the impact of ritonavir-boosted nirmatrelvir (paxlovid) on viral dynamics in the upper respiratory tract. The authors first utilized a viral dynamic model previously developed by the authors. This model was fitted to a rich data sets of longitudinal nasal viral load measurements collected from 1510 individuals in the NBA cohort. It serves as a baseline for simulating viral load trajectories in untreated individuals. Subsequently, a PK viral kinetic model was developed to describe the paxlovid concentrations, and the PK model is combined with the viral dynamic model to estimate the impact of paxlovid. Comparing the viral load predictions from the model for paxlovid with a small set of aggregated viral load data from individuals who were treated with paxlovid, the authors concluded that the in vivo efficacy is 61-fold lower than predicted from in vitro assays. The authors then employed the calibrated model to elucidate clinically observed phenomena such as the drug's failure as post-exposure prophylaxis and the occurrence of viral rebound in a small yet notable fraction of treated individuals. Simulating clinical scenarios, the authors found that 'earlier initiation and shorter treatment duration are key predictors of post-treatment rebound', and 'extension of early symptomatic treatment to 10 days and post-exposure prophylaxis to 15 days is predicted to significantly lower the incidence of viral rebound'. Overall, the work is interesting in that it uses mathematical models to propose potential explanations and mitigation strategies, potentially contributing to the understanding of how paxlovid impacts SARS-CoV-2 kinetics and informing clinical practice.

However, I have the following major concerns about the methodology used to reach these conclusions:

- a) One major weakness of the work is that the number of data points used to calibrate the model for paxlovid is small. While I understand that the parameters in the PK/PD model were calibrated using data from healthy individuals, and the parameters in the viral dynamic model were calibrated using a rich dataset from the NBA cohort, the data used to calibrate the combined model for paxlovid has only a few data points that were aggregated from many individuals. This raises questions about whether the impact of paxlovid can be reliably estimated and how large the uncertainties of the estimates are. The work makes several interesting clinically relevant predictions as listed above. The validity of these predictions depends on the reliability of the estimated impact of paxlovid. I believe these predictions/conclusions would be greatly strengthened if the model for paxlovid is calibrated using individual viral load trajectories from individuals who were treated with paxlovid, and/or if some of these predictions from the model are directly validated by clinical trial data. I think the former is feasible given the availability of clinical data from individuals who were treated with paxlovid.
- b) Because of the aggregated nature of the data used to calibrate the model for paxlovid, the estimated impact of paxlovid represents a population average which ignores the (possibly large) heterogeneity in the impact of the drug on individual level. That is to say, individual-level predictions, such as the histograms of the viral load drop (Fig. 3d, e), and rebound probability (Fig. 4b, 5b, 6c, and 7), are reliable. This again emphasizes the importance of calibrating the model for paxlovid using individual viral load data.
- c) While I appreciate that the authors discuss the limitations of the model in length, another limitation of the model is that the magnitude (parameter m) and timing (parameter τ) of the adaptive immune response are unrelated to the innate immune response or the viral dynamics. However, in reality, the adaptive immune response is likely to be influenced by these factors. In this case, early paxlovid treatment may also impact the two parameters governing adaptive immunity, further impacting viral rebound probabilities. I understand that the choice of the model assumption is reasonable without data, but this possibility should be discussed at least.

Reviewer #2:

Remarks to the Author:

This analysis is an encouraging example how theoretical work can provide further insights into viral-host interactions and how the results can motivate future treatment schedules. The authors present a very elegant approach to link viral dynamics with PKPD models to evaluate the outcomes of different treatment scenarios with ritonavir-boosted nirmatrelvir after an early symptomatic SARS-CoV-2 infection. They investigated onset of treatment, treatment duration, dosage, and dosing frequency to explain the outcomes of one clinical trial. They showed how important the frequency of sampling can be to determine treatment outcomes such as rebounds. Additionally, they determined the in vivo potency of the drug based on data from two clinical trials.

In the following I have listed my comments separated by major and minor comments, which hopefully help to improve the manuscript.

Major comments:

Line 105-109, TableS2, Table S3: I do not understand the differences between the two different described single doses of 250mg/100mg. Why are they different? Could you please add the model fits for the different dosages? And if I compare parameters correctly, the estimated parameters are dose-independent besides the K_{α} (rate of absorption). Please correct this in sentence of line 109. Also FigS2, TableS3 require further explanation for fed and no-fed.

The investigation of dosage differences is particular important as the PK data is based on 250mg/100mg, while the clinical trial data is based on 300mg/100mg. This should be highlighted.

Lines 131/132: Which estimated population distribution exactly? "their" is difficult to allocate in this case. The description of the simulation of the treatment arm is, therefore, difficult to follow.

Line 560: It is not clear for me how you derive start of treatment from the incubation period. Or is this not the case, then please specify how you select start of treatment.

The simulation process requires a more detailed description.

Lines 164ff: How do the results look like for $prf=37$?

Why have you focused mainly on the EPIC-HR trial and not on the PLATCOV trial?

Lines 308ff: Very speculative paragraph. How do you define "destined for more severe infections"?

How do you explain the higher rebound probability in group 3, as it is not really different to group 5. This paragraph requires definitely more explanation and I question if this paragraph is even required here or should be moved completely into the discussion.

Minor comments:

Title and abstract are missing to name the virus.

(Line 18: from the main text I assume we are talking here about a SARS-CoV-2 infection)

Line 46: Which time frame is considered to still call it a viral rebound? And can we exclude that it has been a new infection?

I am a bit confused on how many individuals the viral dynamic model was fitted at the end. The text says 1510, FigS1 shows 130 and TableS1 mentions 589. Could you please clarify?

Lines 92, 102, 126/127: wrong figure reference

Line 97: replace "and" by comma

While the error model used in Monolix is defined for the viral dynamics model in TableS1, I cannot find any equivalent description for the error model used in fitting the PK model. Please specify.

Line 114: Which model exactly are you fitting?

Line 121: Are the individuals with the Omicron infection also from the NBA cohort?

Lines 122-125: Why is symptom onset of importance? Is this not already given for each individual in the NBA cohort? Is not treatment onset more of importance?

Line 211: Are those percentages of rebound observed in the simulations or the clinical trials? Please state clearly.

Lines 250ff: (However, ..., due to ...) Sentence connection does not make sense. Please clarify.

Line 355: Introduction of drug's half-life should already happen in result section while introducing PK model.

Lines 371-373: Link between these two sentences is unclear.

As you have tested now different treatment scenario, which is the most effective one? Prolonged treatment and late treatment initiation seem to result in less frequent rebounds but the untreated case still seems to have the best prospects regarding rebounds. Please discuss timing of therapy in more details.

Why have you decided to perform new simulations for the individual scenarios, instead performing simulations once (See Fig2+3) and then select and sort for the different scenarios? Are all scenarios based on 400 simulations?

Line 460: How are "symptomatic high-risk individuals" defined?

Line 462+477: With which tool/How have you digitized the data?

Line 462: manuscript(1)? Looks like something went wrong with the referencing.

Line 467: "0 through 7" ... what exactly do you mean? Unclear

Line 473: Add (LOD)

Line 475: What is the maximum LOD? Please specify.

Line 479: What means fed and not fed? Unspecific, please specify.

Lines 493ff: pi is not mentioned and explained

Line 518+537: It would be nice to have the Monolix settings (error model, distribution) here or at least as a reference to the tables, so that the method section is complete and compact.

Lines 524: $\gamma=15^{26}$ is huge, isn't it? How do you verify this value?

Lines 547: Are you making use of formula (3) anywhere in the manuscript?

Fig1: Parameter phi is not introduced, please define

Fig1: Virus and plasma volume both have the abbreviation V, please change this to prevent confusion

FigS1: y-axes should capture the same range

FigS1: Please specify which model is used (viral dynamic model)

TableS1: In which terms is the clearance rate of infected cells (I assume delta) density and time-dependent? What does "I" stand for?

TableS2: date is missing for the reference

TableS4: Is $prf=1(?)$ fixed as it is in vitro?

Fig2/3: How are the ranges exactly obtained?

FigS8: How are PEP, early, intermediate, and late defined?

Fig7: Why are 20 days treatment not shown? Please add.

Fig8: I is not introduced under (a) middle row. What is the range on the lower row?

Fig8: Lines 862ff: Sentence is very confusing, please clarify.

Reviewer #3:

Remarks to the Author:

The modeling approach is robust, incorporating both viral dynamics and the effects of an antiviral agent extrapolated from in vitro studies on the delta variant. The in vivo half maximal inhibitory concentration (IC50), adjusted by a correction factor from its in vitro counterpart, is acknowledged, although it derives from a relatively healthier demographic, exclusively male individuals aged 18-30 years (reflecting typical NBA participant profiles). The authors have meticulously detailed their methodology and findings. The central hypothesis - yet to be confirmed through in vivo studies - suggests that post-exposure prophylaxis (PEP) is ineffective, or at least unconvincing, with nirmatrelvir due to the virus potentially evading the adaptive immune system. This is because the antiviral agent rapidly reduces the viral load, possibly hindering an adequate immune response. Notably, a viral rebound is frequently observed in this group post-PEP (though the question remains why not post-early treatment?). The modeling efforts are commendable and potentially applicable to other diseases or future pandemics. Perhaps, the hypothesis concerning the rebound mechanism remains somewhat preliminary at this juncture, especially in the absence of other hypotheses or work.

Review to

A unifying model to explain high nirmatrelvir therapeutic efficacy, low post-exposure prophylaxis efficacy, and frequent viral rebound

Nat Comm

COMMENTS

Abstract: Please specify how early in the symptomatic phase you are referring to. Clarify the clinical trials mentioned by specifying their names or details in the abstract. The statement about a maximally potent agent reducing the viral load by approximately 3.5 logs relative to placebo at 5 days is unclear. Is this agent hypothetical, derived from modeling? This shift in focus seems disconnected from the preceding discussion.

Introduction: The authors propose a model-generated explanation for the failure of antivirals as post-exposure prophylaxis (PEP) in COVID-19 cases. It raises the question of why such an adaptive response does not become blunted with other viruses, such as HIV, where antiretroviral (ARV) drugs work effectively for both PreP and PEP. The suggestion that the virus rebounds after PEP in the absence of antibodies is intriguing. Why is Paxlovid treatment administered for only 5 days when antibodies take about 20 days to develop? Does a viral rebound occur after Paxlovid treatment as well? Despite conventional wisdom advocating for early viral load reduction, the possibility that nirmatrelvir could be effective for PEP in the future should not be dismissed (PEP trials are not trivial to design and analyze).

The phrase "is approximately significantly" needs correction for clarity.

Methods:

The pharmacokinetics (PK) data are derived from a study involving 8 healthy individuals. Given the known variability of nirmatrelvir concentrations, especially after the first dose, it's crucial to understand how the authors modeled the PK of repeated dosing.

Were in vitro pharmacodynamics (PD) from delta strains used to match the NBA dataset? The relevance of IC50 values for new variants and the potential limitations of this approach should be discussed.

The text lacks clarity on which of the 3 parameters are regressed.

What is considered as viral "load"? Details about the measurement of viral load, including its location in the lungs and potential variability due to sampling location, are crucial and should be discussed, including the limit of detection.

Results: Missing references in parentheses indicate a possible typesetting error.

The repetition of methods in the Results section could be reduced.

The most significant finding, regarding the conditions that allow viral rebound after treatment cessation, is intriguing but does not fully explain the absence of similar rebounds with other treatments. The discussion about the possibility of a viral reservoir not targeted by antivirals in the early stages, such as lung epithelium lining, is worth exploring or at least discussed in the Discussion or at the introduction. In other words, why your proposed mechanism should be the one?

The last paragraph appears to be more suited to a Discussion section, as it suggests future directions based on these results and the “parallel” publication.

The variability shown in Fig 2b and the lack of a clear demonstration of rebounds need addressing. The study's limitation by not including antibody maturation and clearance reduces its applicability and contradicts the claim of a "viral-immune" model in the abstract, given the absence of an adaptive immune component.

Discussion: The statement regarding the potential impact of a maximally potent agent on viral load reduction is a highlight of the modeling work. However, the feasibility of achieving such results with a single protease inhibitor (PI) administered orally is questionable. The acknowledgment of the model's limitations aligns with the concerns raised above. The limitation of not having an adaptive immune response modeling (perhaps not easy) is quite a limitation in the application of the model which might be suitable only at early infection stages.

We thank the reviewers for their thoughtful and detailed comments. These recommendations have significantly improved our manuscript. We have tried our best to address all the comments below.

REVIEWER COMMENTS

Reviewer #1 (Remarks to the Author):

Esmaeili-Wellman et al. developed PK/PD and viral dynamic models to understand the impact of ritonavir-boosted nirmatrelvir (paxlovid) on viral dynamics in the upper respiratory tract. The authors first utilized a viral dynamic model previously developed by the authors. This model was fitted to a rich data sets of longitudinal nasal viral load measurements collected from 1510 individuals in the NBA cohort. It serves as a baseline for simulating viral load trajectories in untreated individuals. Subsequently, a PK viral kinetic model was developed to describe the paxlovid concentrations, and the PK model is combined with the viral dynamic model to estimate the impact of paxlovid. Comparing the viral load predictions from the model for paxlovid with a small set of aggregated viral load data from individuals who were treated with paxlovid, the authors concluded that the in vivo efficacy is 61-fold lower than predicted from in vitro assays. The authors then employed the calibrated model to elucidate clinically observed phenomena such as the drug's failure as post-exposure prophylaxis and the occurrence of viral rebound in a small yet notable fraction of treated individuals. Simulating clinical scenarios, the authors found that 'earlier initiation and shorter treatment duration are key predictors of post-treatment rebound', and 'extension of early symptomatic treatment to 10 days and post-exposure prophylaxis to 15 days is predicted to significantly lower the incidence of viral rebound'. Overall, the work is interesting in that it uses mathematical models to propose potential explanations and mitigation strategies, potentially contributing to the understanding of how paxlovid impacts SARS-CoV-2 kinetics and informing clinical practice.

We appreciate this positive feedback.

However, I have the following major concerns about the methodology used to reach these conclusions:

a) One major weakness of the work is that the number of data points used to calibrate the model for paxlovid is small. While I understand that the parameters in the PK/PD model were calibrated using data from healthy individuals, and the parameters in the viral dynamic model were calibrated using a rich dataset from the NBA cohort, the data used to calibrate the combined model for paxlovid has only a few data points that were aggregated from many individuals. This raises questions about whether the impact of paxlovid can be reliably estimated and how large the uncertainties of the estimates are. The work makes several interesting clinically relevant predictions as listed above. The validity of these predictions depends on the reliability of the estimated impact of paxlovid. I believe these predictions/conclusions would be greatly strengthened if the model for paxlovid is

calibrated using individual viral load trajectories from individuals who were treated with paxlovid, and/or if some of these predictions from the model are directly validated by clinical trial data. I think the former is feasible given the availability of clinical data from individuals who were treated with paxlovid.

We thank the reviewer for the comment. We do not completely agree with the statement that there are too few data points for modal validation, particularly for the PLATCOV trial. By comparing the model to the variance, we are incorporating all observed individual data from the trial. Moreover, we are only fitting a single unknown parameter to the mean drop in baseline from multiple time points such that the number of data points exceeds the number of unknown parameters, making unidentifiability unlikely.

However, we agree that fitting to individual data is a good idea that could reinforce our existing conclusions. In the revised version, we refit the model concurrently to individuals with omicron infection from the NBA cohort (to inform viral upslope kinetics which are often not available in the trial participants), as well as to controls and Paxlovid treated study participants in PLATCOV. We used the same unknown parameters as in our published paper on the NBA cohort model (<https://insight.jci.org/articles/view/176286/sd/1>) and added a covariate to account for a longer gap between infection and first quantitative viral load measurement in the control and treatment arm in PLATCOV, and an unknown potency reduction factor for the PLATCOV treatment arm. We achieved excellent model fit while maintaining good parameter identifiability. We obtained a reasonably narrow range of potency reduction factors (prf) for individuals and the mean of our prior model fell within one standard deviation of the prf from the individual fits. Notably the individual fits captured instances of viral rebound in the paxlovid arm. This additional modeling is now included in Figure 3.

To further assess the individual model fitting, we simulated bi-directional counterfactual scenarios, first we simulated treatment for individuals in the control arm, and second we simulated a natural course of infection for all treated individuals. This approach also recapitulated observed reductions in viral loads at all trial timepoints and is included in the supplement. As per the reviewer's suggestion, the model is therefore highly validated against individual data using multiple techniques.

Finally, we incorporated the individual variability of the prf in our simulations and arrived at the same conclusions as in our prior modeling when we reperformed analyses for different doses, dosing intervals, duration of treatments, and timing of treatment initiation in revised versions of Figures 4-7. One difference is that we only used parameter sets from infections with the omicron variant for these updated figures to match PLATCOV. As a result, rebound was less common overall, particularly with extended duration of treatment.

b) Because of the aggregated nature of the data used to calibrate the model for paxlovid, the estimated impact of paxlovid represents a population average which ignores the (possibly large) heterogeneity in the impact of the drug on individual level. That is to say, individual-level predictions, such as the histograms of the viral load drop (Fig. 3d, e), and rebound probability (Fig. 4b, 5b, 6c, and 7), are reliable. This again emphasizes the importance of calibrating the model for paxlovid using individual viral load data.

As mentioned above, we re-simulated the histograms using varying potency reduction factors obtained from individual fits and obtained similar results.

c) While I appreciate that the authors discuss the limitations of the model in length, another limitation of the model is that the magnitude (parameter m) and timing (parameter τ) of the adaptive immune response are unrelated to the innate immune response or the viral dynamics. However, in reality, the adaptive immune response is likely to be influenced by these factors. In this case, early paxlovid treatment may also impact the two parameters governing adaptive immunity, further impacting viral rebound probabilities. I understand that the choice of the model assumption is reasonable without data, but this possibility should be discussed at least.

We agree with this assessment and explored this in our now published manuscript on the NBA cohort (<https://insight.jci.org/articles/view/176286/sd/1>) and identified correlations between the intensity of innate and acquired immune components. Discussion of these findings are beyond the scope of the already very complex current manuscript. Globally speaking, highly granular immune data is needed to verify this model prediction.

Reviewer #2 (Remarks to the Author):

This analysis is an encouraging example how theoretical work can provide further insights into viral-host interactions and how the results can motivate future treatment schedules. The authors present a very elegant approach to link viral dynamics with PKPD models to evaluate the outcomes of different treatment scenarios with ritonavir-boosted nirmatrelvir after an early symptomatic SARS-CoV-2 infection. They investigated onset of treatment, treatment duration, dosage, and dosing frequency to explain the outcomes of one clinical trial. They showed how important the frequency of sampling can be to determine treatment outcomes such as rebounds. Additionally, they determined the in vivo potency of the drug based on data from two clinical trials.

Thank you for the positive feedback.

In the following I have listed my comments separated by major and minor comments, which hopefully help to improve the manuscript.

Major comments:

Line 105-109, TableS2, Table S3: I do not understand the differences between the two different described single doses of 250mg/100mg. Why are they different? Could you please add the model fits for the different dosages? And if I compare parameters correctly, the estimated parameters are dose-independent besides the Kappa_alpha (rate of absorption). Please correct this in sentence of line 109. Also FigS2, TableS3 require further explanation for fed and no-fed.

Thank you for this comment. To obtain the PK parameters we fit the PK model to the available individual-level data for the 250/100mg dose of nirmatrelvir/ritonavir. The parameter values from this modeling part of the study are shared in TableS2.

To investigate the dose-dependency of the PK parameters, we fit the PK model to the mean drug plasma concentration data for two doses 250/100 and 750/100. We had to use average data when exploring the dependency of the parameters on the dose since individual-level data was not available for 750/100 mg dose. These results are shared in Table 3. In the original study, the 250/100mg dose was administered to two groups under fed and fasted conditions. The 750/100 mg study was administered only under fed conditions. We changed the labels to “fed” and “fasted” and added an explanation to the captions of Fig S2 and Table S3.

You’re correct that all the PK parameters are dose-independent. κ_a is also dose-independent as it is the same between 250 (fed) and 750 (fed) groups.

We added the model fits to average data of different doses to the supplementary material (Fig S16)

The investigation of dosage differences is particular important as the PK data is based on 250mg/100mg, while the clinical trial data is based on 300mg/100mg. This should be highlighted.

We thank the reviewer and agree that this is an important point. We have addressed possible uncertainty with dose conversion from 250 to 300 mg by performing a sensitivity analysis of the estimated potency reduction factor sensitivity to possible dose dependent differences in PK parameters (Fig S10 in the revised version) and discussing it in the manuscript, lines 226-230.

Lines 131/132: Which estimated population distribution exactly? “their” is difficult to allocate in this case. The description of the simulation of the treatment arm is, therefore, difficult to follow.

Thank you for this feedback. We reworded this sentence to clarify that the PK parameters were randomly drawn from their respective lognormal population distributions while the PD parameters were from their normal distributions.

Line 560: It is not clear for me how you derive start of treatment from the incubation period. Or is this not the case, then please specify how you select start of treatment. The simulation process requires a more detailed description.

The start of the treatment is determined relative to the symptom onset (as occurred in both trials). Since the time of symptom onset is not available for most of the individuals in the NBA cohort, we first assigned a random incubation period to each selected individual to determine their onset of symptoms. These values were drawn from variant-specific distributions reported in the literature by Gamiche et al. We then drew a delay between symptom onset and treatment start from an appropriate uniform distribution, according to the trial we were simulating. The timing of treatment relative to infection is the sum of the incubation period and the symptom→treatment delay.

Lines 164ff: How do the results look like for prf=37?
Why have you focused mainly on the EPIC-HR trial and not on the PLATCOV trial?

It was a subjective choice to focus on EPIC-HR, based on the larger sample size and the fact that EPIC-HR was the foundational trial demonstrating both virologic and clinical improvements. However, with the further analysis performed in response to reviewer comments, we now focus on both trials more equally. Results are nearly equivalent for slightly different prf. We have elected not to show this in the supplement of the revised version given space limitations. Please note our response to Reviewer 1 in which we re-ran analyses in Figures 4-7 with randomly selected prfs based on individual fits.

Lines 308ff: Very speculative paragraph. How do you define “destined for more severe infections”? How do you explain the higher rebound probability in group 3, as it is not really different to group 5. This paragraph requires definitely more explanation and I question if this paragraph is even required here or should be moved completely into the discussion.

Thank you. We agree that our writing was imprecise here. Our new results with the new prf value estimated for the PLATCOV trial show that individuals with higher peak VL and faster infection expansion (upslope) are more prone to viral rebound after early or PEP treatment. We have generally rephrased this paragraph but chose to keep it to emphasize that host viral-immune kinetics also have bearing on the likelihood of rebound. We do not think it is wholly due to the timing and characteristics of drug treatment.

Minor comments:

Title and abstract are missing to name the virus.

(Line 18: from the main text I assume we are talking here about a SARS-CoV-2 infection)

Thank you for bringing this to our attention. We revised the title and the abstract to include SARS-CoV-2 as the pathogen of interest in this study.

Line 46: Which time frame is considered to still call it a viral rebound? And can we exclude that it has been a new infection?

In the original study, the researchers follow the individuals treated with Paxlovid for 16 days after the start of the treatment. Any re-infection in this time frame is unlikely to be a common occurrence.

I am a bit confused on how many individuals the viral dynamic model was fitted at the end. The text says 1510, FigS1 shows 130 and TableS1 mentions 589. Could you please clarify?

Thank you for bringing this discrepancy to our attention. The viral dynamics model is fit to 1510 infections in the NBA cohort. FigS1 shows 130 sample individuals. 589 in the caption of Table S1 was a typo that is now fixed in the revised version.

Lines 92, 102, 126/127: wrong figure reference

The figure references are all correct. We moved the reference to Fig 1a to a more appropriate place (line 91 in the revised version) to avoid confusion. Other references referencing PK model diagram Fig 1b, and model fits the control arms of both trials (Figs 2a and 3a) are all correct. In the revised version these references are in lines 102, 148, and 149.

Line 97: replace “and” by comma

Thank you for your attention to detail. We made the change.

While the error model used in Monolix is defined for the viral dynamics model in TableS1, I cannot find any equivalent description for the error model used in fitting the PK model. Please specify.

Thank you for bringing this to our attention. We used Monolix’s proportional error model. This and the estimated error parameter are now mentioned in the caption of Table S2.

Line 114: Which model exactly are you fitting?

We fit the simulation results of the viral dynamics+PKPD models to the aggregate trial data. We reworded the sentence to be more clear (lines 114-120).

Line 121: Are the individuals with the Omicron infection also from the NBA cohort?

Yes. All the individuals in the virtual cohort are from the NBA cohort. We reworded the sentence (lines 129-131)

Lines 122-125: Why is symptom onset of importance? Is this not already given for each individual in the NBA cohort? Is not treatment onset more of importance?

As we explained above, the treatment start day is relative to the time of symptom onset. The timing of symptom onset is unknown for most individuals in the NBA cohort so we had to randomly select incubation period, and duration of symptoms at enrollment for each simulated individual.

Line 211: Are those percentages of rebound observed in the simulations or the clinical trials? Please state clearly.

Thank you for this clarifying question. All the percentages are obtained from the simulation. We clarified this in the text.

Lines 250ff: (However, ..., due to ...) Sentence connection does not make sense. Please clarify.

We apologize for the lack of clarity. This paragraph is revised with new results in the new version and we tried to avoid long sentences to avoid confusion.

Line 355: Introduction of drug's half-life should already happen in result section while introducing PK model.

We moved the half-life introduction to the line 213 where it's first discussed.

Lines 371-373: Link between these two sentences is unclear.

Both sentences discuss the suspected risk factors for a viral rebound. We slightly reworded the last sentence to make the connection more clear.

As you have tested now different treatment scenario, which is the most effective one? Prolonged treatment and late treatment initiation seem to result in less frequent rebounds but the untreated case still seems to have the best prospects regarding rebounds. Please discuss timing of therapy in more details.

Early treatment is important to prevent hospitalization and death, especially in high-risk populations. Based on published studies cited in the paper, lower rebound probability in the control group and late treatment group doesn't justify not treating or treating late. According to our model, for vaccinated individuals infected by current VOC (Omicron), an option that should be formally tested is to treat early and extend the treatment to 10 days. This is discussed in multiple places in the manuscript, including the discussion.

Why have you decided to perform new simulations for the individual scenarios, instead performing simulations once (See Fig2+3) and then select and sort for the different scenarios? Are all scenarios based on 400 simulations?

Figures 2 and 3 show the simulation of the treatment regimen used in these trials (300 mg, twice per day, starting within 3-4 days after symptom onset, for 5 days). When we explore other scenarios, we change dosing, frequency, timing, and duration one at a time. Results for each treatment regimen are based on the simulation of the combined model for 400 individuals randomly selected from the NBA cohort.

Line 460: How are “symptomatic high-risk individuals” defined?

In the original study by Hammond et al., high-risk is defined as having at least one risk factor for hospitalization or death.

Line 462+477: With which tool/How have you digitized the data?

Thank you for this question. We used web plot digitizer (an online tool) that can be accessed here: <https://automeris.io/WebPlotDigitizer.html>.

We added this explanation and citation to the manuscript (line 520)

Line 462: manuscript(1)? Looks like something went wrong with the referencing.

Thank you for bringing this to our attention. There was a mistake with the citation format which is now fixed.

Line 467: “0 through 7” ... what exactly do you mean? Unclear

The samples in the PLATCOV study were collected daily on days 0 through 7 and day 14 after treatment start day. We added the word “daily” to clarify.

Line 473: Add (LOD)

Line 475: What is the maximum LOD? Please specify.

Thank you for pointing this out. We added the LOD values for both trials.

Line 479: What means fed and not fed? Unspecific, please specify.

The categories fed and not fed refer to individuals in the original study by Singh et al. who took the medication in the fed and fasting conditions, respectively. We changed not fed to fasting for clarity.

Lines 493ff: pi is not mentioned and explained

Thank you for bringing this to our attention. We added a phrase to introduce pi. (line 543)

Line 518+537: It would be nice to have the Monolix settings (error model, distribution) here or at least as a reference to the tables, so that the method section is complete and compact.

Thank you for this comment. We added a reference to the appropriate tables for the parameter distributions and error model.

Lines 524: $\gamma=15^{26}$ is huge, isn't it? How do you verify this value?

We apologize for this confusion. Gamma = 15. The superscript 26 is the citation following nature communication's citation format.

Lines 547: Are you making use of formula (3) anywhere in the manuscript?

Yes. Equation 3 is used to calculate the average efficacy mentioned in line 214. We added a reference to equation 3 to clarify.

Fig1: Parameter phi is not introduced, please define

Thank you for bringing it to our attention. We added the definition to the caption.

Fig1: Virus and plasma volume both have the abbreviation V, please change this to prevent confusion

We appreciate your attention to detail. We changed the plasma volume to Vol.

FigS1: y-axes should capture the same range

We made all the y-axis the same range.

FigS1: Please specify which model is used (viral dynamic model)

Thank you. We updated the caption.

TableS1: In which terms is the clearance rate of infected cells (I assume delta) density and time-dependent? What does “I” stand for?

Thank you for bringing this to our attention. This was a typo and is fixed in the revised manuscript.

TableS2: date is missing for the reference

We apologize. The date is now added to the reference.

TableS4: Is $prf=1$ (?) fixed as it is in vitro?

Yes. The PD parameters are estimated from in vitro data which assumes $prf = 1$.

Fig2/3: How are the ranges exactly obtained?

The range of prf in figures 2 and 3 are obtained by running the simulations 10 times as it's explained in the caption of both figures and the manuscript (lines 165-167)

FigS8: How are PEP, early, intermediate, and late defined?

We added the definitions to the caption.

Fig7: Why are 20 days treatment not shown? Please add.

20 days treatment is now shown in the updated version.

Fig8: I is not introduced under (a) middle row. What is the range on the lower row?

Thank you for your attention. We added “I” introduction to the caption. The bands on the lower row show the 95% confidence interval.

Fig8: Lines 862ff: Sentence is very confusing, please clarify.

Thank you for this feedback. We reworded the sentence to clarify.

Reviewer #2 (Remarks on code availability):

I have had a quick look over the code availability, and can confirm that the code and supplementary datasets are available, also a README is provided. The code looks well maintained, but I would not go so far that I would say I have really reviewed the code.

Reviewer #3 (Remarks to the Author): Also, see attached document

The modeling approach is robust, incorporating both viral dynamics and the effects of an antiviral agent extrapolated from in vitro studies on the delta variant. The in vivo half maximal inhibitory concentration (IC50), adjusted by a correction factor from its in vitro counterpart, is acknowledged, although it derives from a relatively healthier demographic, exclusively male individuals aged 18-30 years (reflecting typical NBA participant profiles). The authors have meticulously detailed their methodology and findings. The central hypothesis - yet to be confirmed through in vivo studies - suggests that post-exposure prophylaxis (PEP) is ineffective, or at least unconvincing, with nirmatrelvir due to the virus potentially evading the adaptive immune system. This is because the antiviral agent rapidly reduces the viral load, possibly hindering an adequate immune response. Notably, a viral rebound is frequently observed in this group post-PEP (though the question remains why not post-early treatment?). The modeling efforts are commendable and potentially applicable to other diseases or future pandemics. Perhaps, the hypothesis concerning the rebound mechanism remains somewhat preliminary at this juncture, especially in the absence of other hypotheses or work.

Thank you so much for the positive feedback. We feel that the hypotheses of mechanisms underlying viral rebound have been at least somewhat validated, based on several factors already mentioned in our study. First, in animal models of infection, early paxlovid does appear to suppress early immune responses. Second, monoclonal antibodies with long half-life work as post-exposure prophylaxis and are not associated with viral rebound, as predicted by our model. Third, viral rebound occurs with less virologically potent, short half-life agents such as molnupiravir and remdesivir though less commonly in some studies, which aligns with our prediction that higher potency short half-life agents increase the probability of rebound.

COMMENTS

Abstract: Please specify how early in the symptomatic phase you are referring to. Clarify the clinical trials mentioned by specifying their names or details in the abstract.

Thank you so much for this comment. We added the details about timing of treatment in the EPIC-HR trial and the name of trials to the abstract.

The statement about a maximally potent agent reducing the viral load by approximately 3.5 logs relative to placebo at 5 days is unclear. Is this agent hypothetical, derived from modeling? This shift in focus seems disconnected from the preceding discussion.

Yes. This statement is based on our modeling prediction of the maximally potent drug. We added a phrase clarifying this.

Introduction: The authors propose a model-generated explanation for the failure of antivirals as post-exposure prophylaxis (PEP) in COVID-19 cases. It raises the question of why such an adaptive response does not become blunted with other viruses, such as HIV, where antiretroviral (ARV) drugs work effectively for both PreP and PEP. The suggestion that the virus rebounds after PEP in the absence of antibodies is intriguing. Why is Paxlovid treatment administered for only 5 days when antibodies take about 20 days to develop? Does a viral rebound occur after Paxlovid treatment as well? Despite conventional wisdom advocating for early viral load reduction, the possibility that nirmatrelvir could be effective for PEP in the future should not be dismissed (PEP trials are not trivial to design and analyze).

The phrase "is approximately significantly" needs correction for clarity.

We apologize for the confusion. The word "approximately" is removed in the revised version.

Methods:

The pharmacokinetics (PK) data are derived from a study involving 8 healthy individuals. Given the known variability of nirmatrelvir concentrations, especially after the first dose, it's crucial to understand how the authors modeled the PK of repeated dosing.

In the PK simulations, for repeated dosing, The relevant dose in each scenario was added to the AGI compartment (the absorption equation) of the PK model (eq 2a) at each dosing timepoint (t=0, 0.5, 1, 1.5, ..., 4.5 days). We added this explanation to the method. (lines 627-630)

Were in vitro pharmacodynamics (PD) from delta strains used to match the NBA dataset? The relevance of IC50 values for new variants and the potential limitations of this approach should be discussed.

Thank you for this question. Yes. While the PD parameters estimated from in vitro delta strain were used in all simulations, the true in vivo value of IC50 for any strain can only be determined using the methods described in this paper. In fact, the difference in Prf values of the two trials could also reflect the potential difference between IC50 of different strains. We clarified this in the discussion in the revised version (lines 172 and 384)

The text lacks clarity on which of the 3 parameters are regressed.

We apologize for the lack of clarity. The 3 parameters are the 3 PD parameters (Emax, hill coefficient, and IC50). We add this explanation to the text (line 589).

What is considered as viral "load"? Details about the measurement of viral load, including its location in the lungs and potential variability due to sampling location, are crucial and should be discussed, including the limit of detection.

The two trials used two sampling methods. In the EPIC-HR trial, samples were measured using nasal swabs, while in the PLATCOV trial, oral swabs were used. In both studies, the viral load was measured using qPCR. We mention this in the methods section (lines 500, 505-506). The limit of detection of each study is also mentioned at the bottom of the same paragraph (lines 512-517)

The difference in the estimated prf for the two trials can be partly attributed to the different sampling methods. We discuss this in the paper (lines 169 and 384)

Results: Missing references in parentheses indicate a possible typesetting error.

We apologize for the typos. We made sure they are fixed in the revised version.

The repetition of methods in the Results section could be reduced.

Thank you for this feedback. We elected to keep some explanation of the methods in the result section to help those readers who are only interested in the results section.

The most significant finding, regarding the conditions that allow viral rebound after treatment cessation, is intriguing but does not fully explain the absence of similar rebounds with other treatments.

Rebound is not limited to Paxlovid. As we discuss in the introduction (line 51), rebound is also observed after other antiviral treatments like Molnupiravir.

The discussion about the possibility of a viral reservoir not targeted by antivirals in the early stages, such as lung epithelium lining, is worth exploring or at least discussed in the Discussion or at the introduction. In other words, why your proposed mechanism should be the one?

Our paper addresses the failure of treatment to completely clear the infection as part of the proposed mechanism of rebound (line 345 and 397). Our proposed mechanism of rebound includes the failure of treatment to completely clear the infection, blunting the early immune response, and preserving susceptible cells.

The last paragraph appears to be more suited to a Discussion section, as it suggests future directions based on these results and the “parallel” publication.

Thank you for this suggestion. We elected to keep this paragraph in the result section as it discussed the potential role of host viral-immune dynamics in the rebound probability in addition to the timing and duration of treatment.

The variability shown in Fig 2b and the lack of a clear demonstration of rebounds need addressing.

Fig 2b shows the individual and mean data and the viral load drop from the baseline of simulated individuals (grey lines). Viral load rebound can be seen in the viral load of some of the simulated individuals increasing after the end of the treatment. In the revised version, we also include sample model fits to individual viral load data of the control and treatment arms of PLATCOV trial. Treatment-induced viral load rebound can be seen more clearly in some of the individuals in the treatment arm (Fig 2g).

The study's limitation by not including antibody maturation and clearance reduces its applicability and contradicts the claim of a "viral-immune" model in the abstract, given the absence of an adaptive immune component.

We discuss this limitation in the discussion (lines 443-453).

Discussion: The statement regarding the potential impact of a maximally potent agent on viral load reduction is a highlight of the modeling work. However, the feasibility of achieving such results with a single protease inhibitor (PI) administered orally is questionable. The acknowledgment of the model's limitations aligns with the concerns raised above. The limitation of not having an adaptive immune response modeling (perhaps not easy) is quite a limitation in the application of the model which might be suitable only at early infection stages.

We agree that, ideally, when immune data is available, modeling adaptive immunity would provide more insight into the roles that different arms of immune response play in clearing the virus. However, in the absence of immune data, any attempt to model the adaptive immune response would be conjecture and not supported by data.

Reviewer #3 (Remarks on code availability):

I do not have that (costly) software to review the model

Reviewers' Comments:

Reviewer #1:

Remarks to the Author:

My concerns are addressed and I do not have further concerns.

Reviewer #2:

Remarks to the Author:

The authors have addressed all my previous comments. In the following I only have a few minor remarks:

Line 93: Add abbreviation (NBA)

Line 102 and 148/149: Something went again wrong with the figure reference

Line 241/242: As these percentages vary from the previous version, I would prefer to see a range or 95% confidence interval given. If it is not too much off a hassle, could you please add them?

FigS16 is wrongly labeled.

I do not consider 250 fed in Fig S16 as a good fit, as the peak is not well captured. Could you improve on this? Or do you have any explanation why the peak is not well captured?

Table S1: Using the viral dynamic model, we estimated ... (without with)

Reviewer #3:

None

REVIEWERS' COMMENTS

Reviewer #1 (Remarks to the Author):

My concerns are addressed and I do not have further concerns.

We appreciate the reviewer's constructive and positive feedback and are glad we could address them all.

Reviewer #2 (Remarks to the Author):

The authors have addressed all my previous comments. In the following I only have a few minor remarks:

We are glad we could address all of the comments. We appreciate your attention to detail.

Line 93: Add abbreviation (NBA)

Thank you for bringing this to our attention. We added the abbreviation.

Line 102 and 148/149: Something went again wrong with the figure reference

We apologize for this issue. The figure references in the version we submitted are correct. We will ensure they remain accurate when we proofread the final version.

Line 241/242: As these percentages vary from the previous version, I would prefer to see a range or 95% confidence interval given. If it is not too much of a hassle, could you please add them?

Of course! We added the 95% CI to the reported percentages.

However, please note that the percentages changed from the first version because we used the cohort characteristics compatible with the PLATCOV trial (vaccinated omicron-infected) in the second version for our simulations (line 233). The rebound probabilities are overall lower in the vaccinated individuals with omicron infection compared to unvaccinated pre-omicron individuals. We discuss this in the paper (line 314)

FigS16 is wrongly labeled.

Thank you! We fixed the label.

I do not consider 250 fed in Fig S16 as a good fit, as the peak is not well captured. Could you improve on this? Or do you have any explanation why the peak is not well captured?

We tried to improve the fit to 250 fed data but still underestimated the two data points. We don't know precisely why our PK model doesn't capture the peak for the fed group.

The peak in the fed groups occurs later than in the fasting group. This may be due to data digitization errors, or a mechanistic difference in the absorption process between the fed and fasting groups. An exploration of a more complex PK model with a different absorption mechanism is beyond the scope of this study. Our model captures the clearance phase of the drug's plasma concentrations well, which, according to our sensitivity analysis, is most important for estimating the potency reduction factor.

Also, we noticed that the label of the 750mg group needed to be corrected. The 750mg group is a fasting group. We sincerely apologize for this typo. We fixed the label in Fig S16 and Table S3. This also suggests that the absorption rate (k_a) might not be dose-independent, but it's inconclusive. The variability in the k_a value between 250 fed, fasting, and 750 fasting is also reflected in the parameter's estimated individual values for 250 fed and fasting groups. We explained this in the caption of Table S3. This does not impact the analysis and results of the main paper. The PK individual parameters are available in our GitHub.

Table S1: Using the viral dynamic model, we estimated ... (without with)

Thank you for noticing this typo. We fixed it.